# Global evaluation of current and future threats to drylands and their vertebrate biodiversity

Amir Lewin [1,2] ✉, Gopal Murali [1,2,3], Shimon Rachmilevitch [4] & Uri Roll [2]

Drylands are often overlooked in broad conservation frameworks and development priorities and face increasing threats from human activities. Here we evaluated the formal degree of protection of global drylands, their land vertebrate biodiversity and current threats, and projected human-induced land-use changes to drylands under different future climate change and socioeconomic scenarios. Overall, drylands have lower protected-area coverage (12%) compared to non-drylands (21%). Consequently, most dryland vertebrates including many endemic and narrow-ranging species are inadequately protected (0–2% range coverage). Dryland vertebrates are threatened by varied anthropogenic factors— including agricultural and infrastructure development (that is, artificial structures, surfaces, roads and industrial sites). Alarmingly, by 2100 drylands are projected to experience some degree of land conversion in 95–100% of their current natural habitat due to urban, agricultural and alternative energy expansion. This loss of undisturbed dryland regions is expected across different socioeconomic pathways, even under optimistic scenarios characterized by progressive climate policies and moderate socioeconomic trends.

Global conservation efforts have made substantial gains in the past 20 years, aided by numerous large-scale scientific examinations and recommendations[1–4]. Much scientific and conservation efforts have focused on the megadiverse tropics[5,6] while neglecting other important biomes[7,8]. Drylands encompass nearly half of the earth's terrestrial surface, support vital ecosystems and endemic species, and provide key ecosystem services to a third of the global human population[9,10]. However, drylands are mostly absent from many dedicated global conservation initiatives[11,12].

Drylands are especially sensitive to increasing human land-use pressures through land conversion due to extensive agriculture and alternative energy sources, overgrazing, invasive species and climate change—pressures that are expected to intensify in the coming decades[13–17]. Human pressures in drylands may result in severe land degradation and diminished overall productivity, threatening dryland ecosystems and biodiversity[18]. It is estimated that up to 20% of drylands are already degraded or at risk[9,18], including large increases in bare ground and lost vegetation cover[15] and considerable water declines in natural lakes, reservoirs[19] and groundwater[20]. Drylands are also sensitive to the introduction and spread of non-native species supported by exogenous resources due to agricultural and other human land uses in otherwise low-productivity desert ecosystems, with cascading

[1]Jacob Blaustein Center for Scientific Cooperation, The Jacob Blaustein Institutes for Desert Research, Ben-Gurion University of the Negev, Midreshet Ben-Gurion, Israel. [2]Mitrani Department of Desert Ecology, The Swiss Institute for Dryland Environmental and Energy Research, The Jacob Blaustein Institutes for Desert Research, Ben-Gurion University of the Negev, Midreshet Ben-Gurion, Israel. [3]Department of Ecology and Evolutionary Biology, University of Arizona, Tucson, AZ, USA. [4]French Associates Institute for Agriculture and Biotechnology of Drylands, Jacob Blaustein Institutes for Desert Research, Ben-Gurion University of the Negev, Midreshet Ben-Gurion, Israel. ✉e-mail: amirlewin@gmail.com

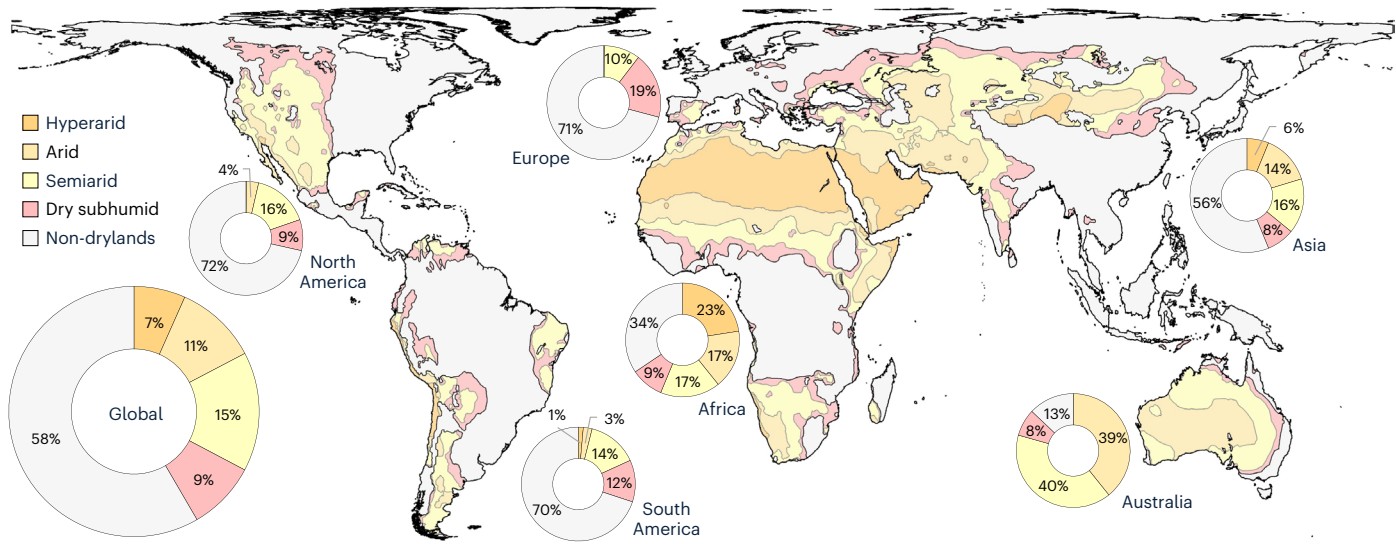

**Fig. 1 | Global drylands and subtypes (UNEP-WCMC)**[31]**.** Circles show the relative terrestrial proportion of dryland subtypes globally and by continent.

**Table 1 | Geographic extent of drylands by subtype and continent**

| | Subtype area (km²) | | | | Total area (km²) | Continent (%) | Drylands (%) |
|---|---|---|---|---|---|---|---|
| | **Hyperarid** | **Arid** | **Semiarid** | **Dry subhumid** | | | |
| Africa | 6,729,849 | 4,994,021 | 5,090,644 | 2,788,943 | 19,603,457 | 66% | 32% |
| Asia | 2,766,902 | 6,341,385 | 6,999,267 | 3,539,180 | 19,646,733 | 44% | 32% |
| Australia | 0 | 3,013,490 | 3,085,886 | 595,663 | 6,695,039 | 87% | 11% |
| Europe | 0 | 32,477 | 1,002,994 | 1,863,490 | 2,898,961 | 29% | 5% |
| North America | 26,559 | 842,281 | 3,914,740 | 2,143,927 | 6,927,507 | 29% | 11% |
| South America | 250,190 | 434,845 | 2,531,392 | 2,100,449 | 5,316,877 | 30% | 9% |
| Total | 9,773,500 | 15,658,499 | 22,624,924 | 13,031,652 | 61,088,574 | | |

ecological effects[21–23]. Greening initiatives and afforestation (for example, the Great Green Wall Initiative) have the potential to exacerbate these drivers through habitat and resource changes[24–26]. Dryland species are also among the most vulnerable to projected global climate changes resulting in extreme heat events and increased aridity that can overwhelm species already at their physiological tolerance limits to heat and water stress[27–29]. Climate change impacts may further be exacerbated in drylands by reduced opportunities for species to adapt and survive due to accelerated habitat loss and fragmentation[13].

Consequently, there is an urgent need for a systematic approach to evaluate current and future threats to drylands and their vertebrate biodiversity. Such an approach will have important implications for guiding land management and conservation strategies in drylands by identifying broad-scale conservation priorities[30] and increasing conservation targets across all ecosystems. Here we assess the degree of protected area coverage in drylands and dryland subtypes compared to non-drylands, the current status of anthropogenic threats to dryland vertebrate biodiversity—highlighting important protection gaps—and the impact that projected future human land-use pressures will have on drylands under different climate and socioeconomic pathways.

## Global drylands

We evaluated drylands as classified by UNEP-WCMC (United Nations Environment World Conservation Monitoring Centre)[31]. Drylands are categorized using an aridity index (the ratio of annual precipitation to potential evapotranspiration), with values below 0.65. Drylands are further divided into hyperarid, arid, semiarid and dry subhumid subtypes (Fig. 1). This drylands designation dataset is commonly referred to by the Convention of Biological Diversity (https://www.cbd.int/gbf/) and the United Nations Convention to Combat Desertification (https://www.unccd.int/) for policy and management goals and often cited in the scientific literature (for example, ref. 10). Drylands cover about 42% of earth's terrestrial surface and encompass diverse and unique regions globally (Fig. 1). Most dryland regions are found in Africa and Asia, which combined encompass 64% of global drylands and also comprise the largest areas of hyperarid and arid subtypes (Table 1). Drylands are mostly inclusive of the deserts and xeric shrublands biome[1,2] and also comprise large proportions of other biomes and diverse habitats[9] across continents. These range from subhumid zones (for example, tropical and subtropical grasslands, savannas and shrublands) especially in regions of Africa and Australia (Fig. 1 and Extended Data Fig. 1) to Mediterranean regions of Europe, temperate zones (for example, temperate grasslands, savannas and shrublands) especially in Asia and North America and small portions of sub-polar regions (for example, boreal forests/taiga and temperate conifer forests).

## Protected areas in drylands

Drylands are considerably less covered by the global network of protected areas compared to non-drylands (Fig. 2a). Using the World Database on Protected Areas (WDPA)[32] and OpenStreetMap[33], we show that only ~12% of total drylands are covered by protected areas (considering all International Union for Conservation of Nature (IUCN)

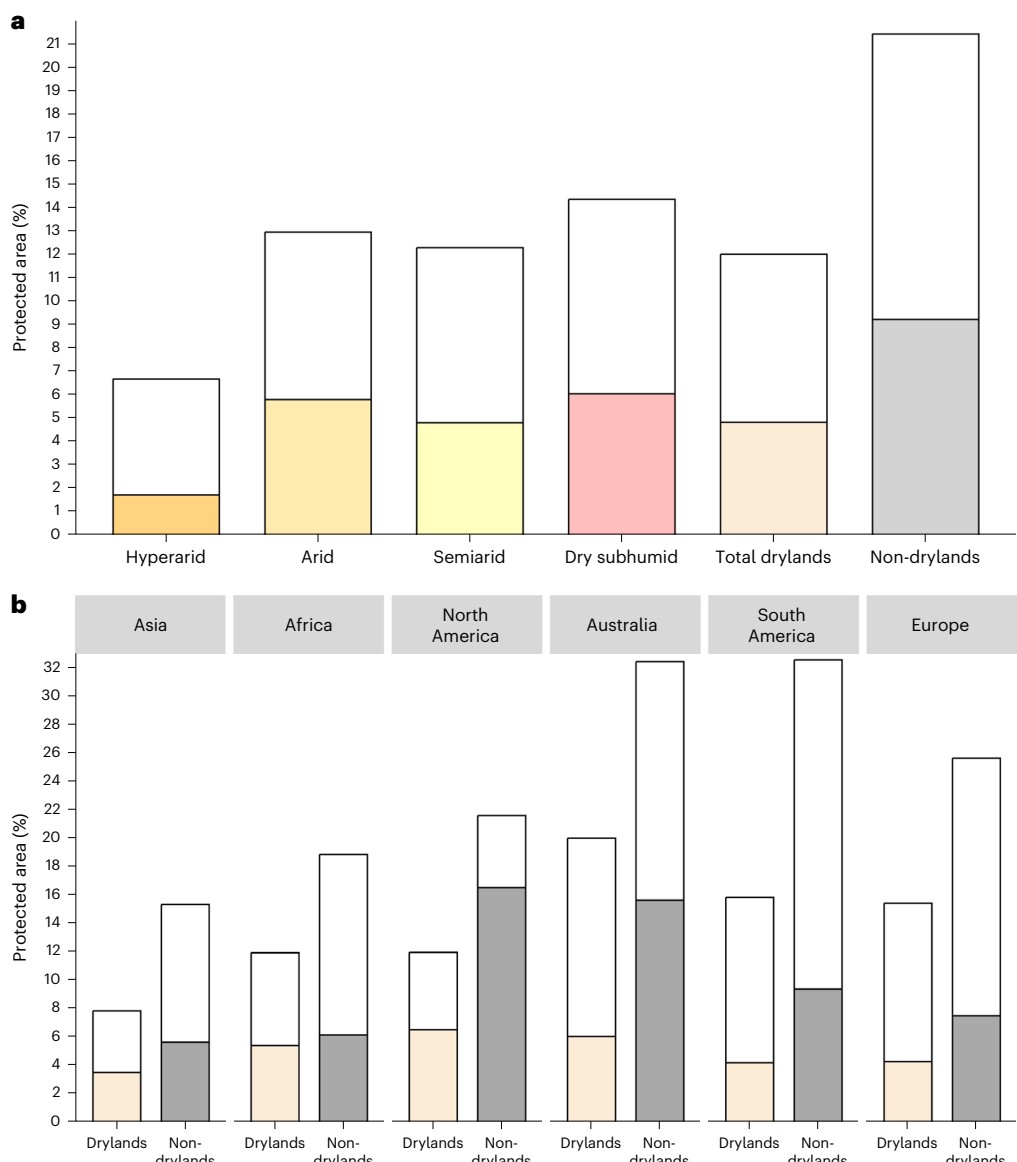

**Fig. 2 | Protected areas in drylands. a**, Proportion of protected areas by dryland subtype compared to total non-drylands. **b**, Proportion of protected areas in drylands by continent compared to non-drylands. Coloured bars, IUCN categories I–IV; white bars, IUCN categories V and VI and uncategorized (that is, 'not applicable', 'not assigned' and 'not reported').

categories and uncategorized protected areas)—well below global conservation targets to be achieved by 2030 (that is, 30% coverage of ecologically representative areas). By contrast, non-dryland regions are better protected with ~21% coverage (Fig. 2a). Some of the largest protected areas in the world are in hyperarid deserts (for example, Uruq Bani Ma'arid in Saudi Arabia), presumably due to low human populations and few competing land uses[34,35]. However, overall the most arid dryland regions, which are home to unique and diverse plant and animal species[9,13], are the least protected (Fig. 2a and Extended Data Fig. 2). For example, dry subhumid regions are 14.4% protected compared to hyperarid regions with only 6.7% protection. Furthermore, different biomes within drylands receive different levels of protection (Extended Data Fig. 3). For example, grasslands and savannahs are among the most biodiverse dryland ecosystems, vulnerable to over-grazing[36] and other impacts, and least protected[30]. Considering protected areas managed mainly for science, wilderness protection and habitat and species conservation (that is, IUCN categories I–IV)[37]— a considerably smaller proportion of drylands are protected. Less than

5% of all dryland subtypes are found in protected areas designated as categories I–IV (Fig. 2a). These patterns remain consistent across continents (Fig. 2b). Such gaps in protected area coverage of drylands have important implications in Africa and Asia, which contain most global drylands (Tables 1 and 2). In Africa and Asia, protected areas including community and indigenous conserved areas without an IUCN category may be under-reported[38], which might actively support effective biodiversity conservation[39]. Conversely, IUCN-designated protected areas may remain unprotected due to poor enforcement[1]. Nevertheless, all continents have a lower coverage of protected areas in categories I–IV (Fig. 2b). For example, in Australian drylands protection decreases from 20% to 6% coverage under IUCN categories I–IV and to less than 7% protection of North American drylands in IUCN categories I–IV. These findings are concerning, as protected areas permitting multiple land uses and resource extraction in drylands (that is, IUCN categories V–VI and uncategorized) might be particularly ineffective in biodiversity conservation due to the sensitivity of fragile dryland ecosystems and species to anthropogenic activities

**Table 2 | Protected areas in drylands by subtype and continent**

| | Subtype area protected IUCN I–IV (km²) | | | | Subtype area protected IUCN V–VI (km²) | | | |
|---|---|---|---|---|---|---|---|---|
| | **Hyperarid** | **Arid** | **Semiarid** | **Dry subhumid** | **Hyperarid** | **Arid** | **Semiarid** | **Dry subhumid** |
| Africa | 124,467 | 277,514 | 371,742 | 291,268 | 269,997 | 166,336 | 487,291 | 351,991 |
| Asia | 36,318 | 281,097 | 260,824 | 115,616 | 202,337 | 334,698 | 245,123 | 70,075 |
| Australia | 0 | 217,289 | 148,640 | 39,744 | 0 | 489,580 | 390,363 | 53,007 |
| Europe | 0 | 1,281 | 43,392 | 80,398 | 0 | 277 | 109,592 | 212,305 |
| North America | 7,563 | 124,507 | 166,546 | 154,357 | 5,173 | 86,146 | 184,287 | 100,463 |
| South America | 2,957 | 9,401 | 103,052 | 108,742 | 5,267 | 42,869 | 273,872 | 294,786 |
| Total | 171,304 | 911,089 | 1,094,196 | 790,125 | 482,774 | 1,119,906 | 1,690,527 | 1,082,626 |

This classification follows IUCN categories I–IV and categories V and VI (which includes uncategorized protected areas designated as: 'not applicable', 'not assigned' and 'not reported').

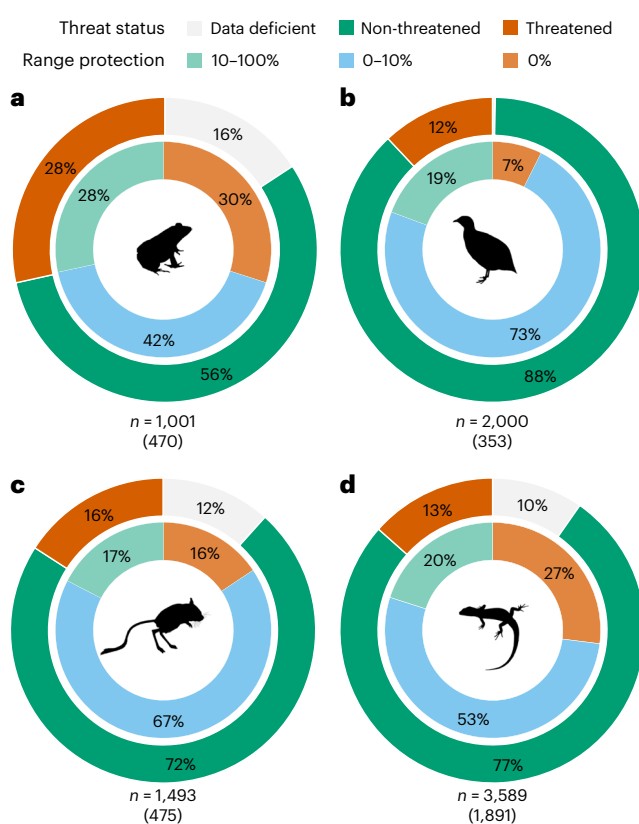

**Fig. 3 | Range size protection and threat status of terrestrial vertebrate species in drylands. a–d,** Amphibian (**a**), bird (**b**), mammal (**c**) and reptile (**d**) species in drylands (≥50% global range in drylands). Inner circles show proportion of species with range size protected (IUCN categories I–IV): orange, 0% range protected; blue, 0–10% range protected; green, 10–100% range protected. Outer circles show proportion of IUCN threatened species: dark orange, threatened (IUCN categories CR, EN and VU); dark green, non-threatened (IUCN categories Near Threatened (NT) and Least Concern (LC)); grey, data deficient (IUCN category Data Deficient (DD)); *n*, number of species; parentheses, number of endemic species (>99% range in drylands).

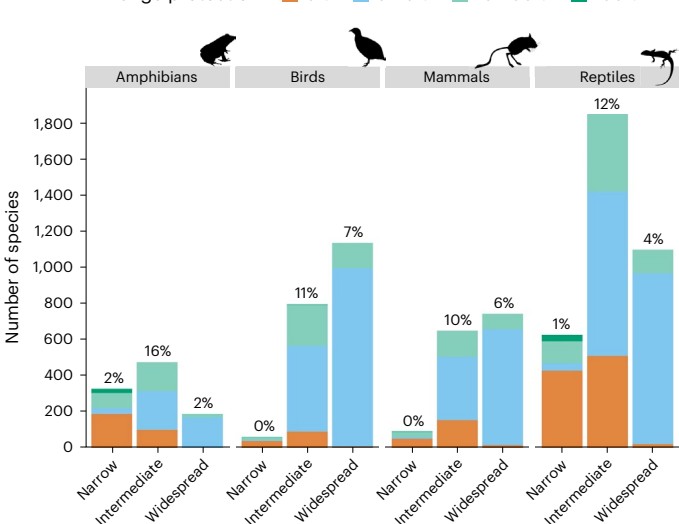

**Fig. 4 | Protection of amphibian, bird, mammal and reptile dryland species (≥50% global range in drylands) according to representation targets.** Narrow-ranging species = <1,000 km²; intermediate = 1,000–250,000 km²; widespread = >250,000 km². The protection target for narrow-ranging species is 100% of range size, whereas the protection target for intermediate and widespread species is 10–100% of range size. Scale shows proportion of species with range size protection (IUCN categories I–IV): orange, 0%; blue, 0–10%; green, 10–100%; dark green, 100% range protection. Percentage values above the bar plots are the fraction of each taxon's protection target covered in protected areas.

and threats[6,40,41] (see 'Threats to dryland biodiversity and conservation gaps' and 'Current and future land-use change scenarios'). Altogether, in drylands we find a disproportionate degree of low protection coverage overall, under-representation in the coverage of important dryland subtypes and biomes, and a low proportion of protected areas designated for biodiversity conservation. These factors are likely to have important implications for the conservation of dryland species and habitats.

## Threats to dryland biodiversity and conservation gaps

Drylands contain a rich diversity of unique and endemic plant and animal species as a result of wide-ranging adaptations to extreme conditions over heterogeneous habitats[9,13,42]. We evaluated the distribution and threat status of terrestrial vertebrate species—amphibian, bird, mammal and reptile species with 50% or more of their global ranges in drylands. We find that the vast majority of dryland species have less than 10% of their ranges protected (considering IUCN protected area categories I–IV)[37] in drylands (Fig. 3 and Supplementary Data 1), which is likely insufficient habitat for maintaining viable populations[14,43]. Worse still, 30% of amphibians, 7% of birds, 16% of mammals and 27% of reptiles have no overlap with protected areas (that is, 'gap species'; Fig. 3, inner circles). Of these species with zero protection in drylands, 224 amphibian, 73 bird, 144 mammal and 720 reptile species are endemic to drylands (that is, species with more than 99% of their global ranges in drylands; Supplementary Data 1). Reptiles are the most diverse of the

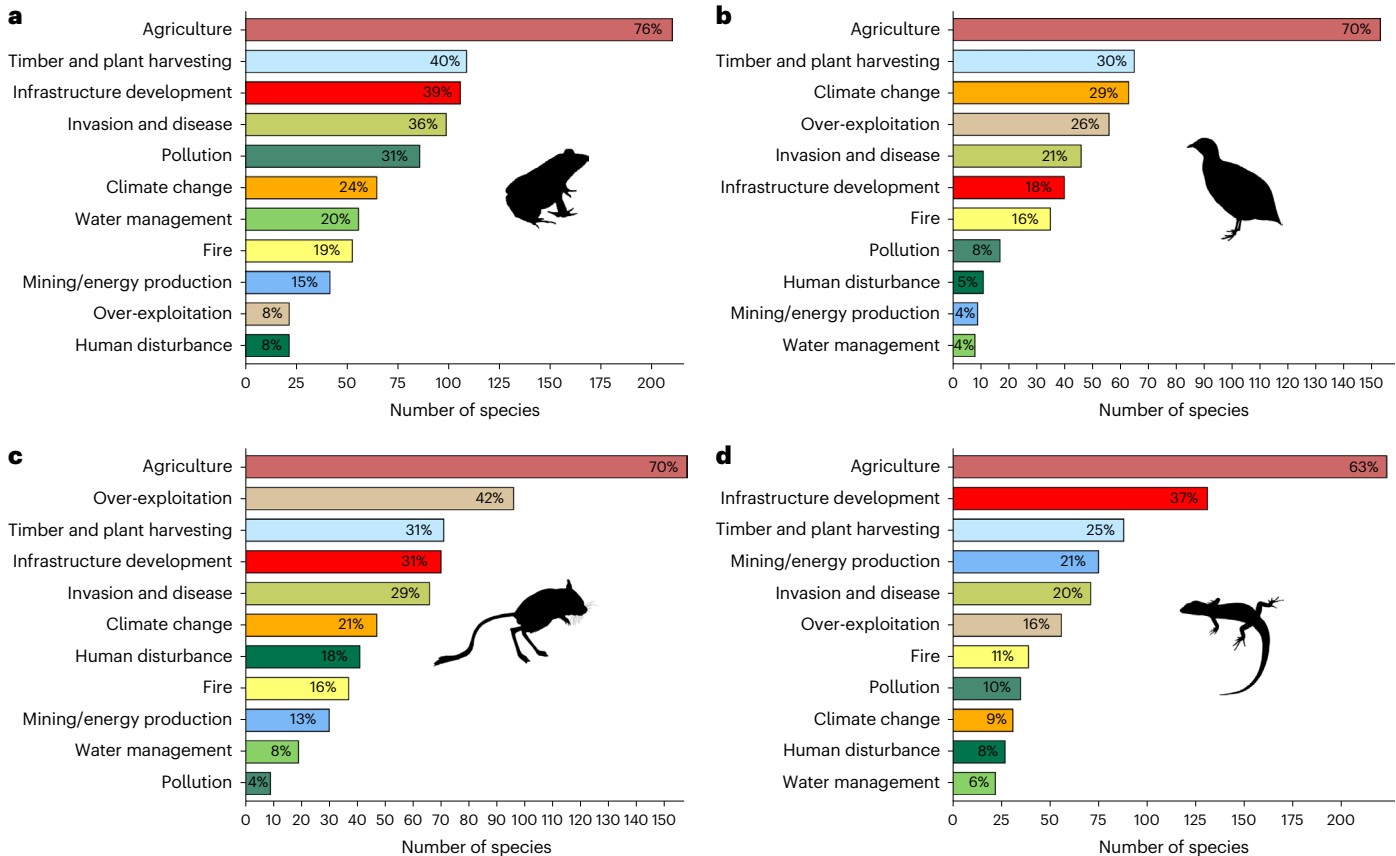

**Fig. 5 | Threats to terrestrial vertebrate species in drylands. a–d**, Amphibian (**a**), bird (**b**), mammal (**c**) and reptile (**d**) threatened species (in IUCN categories CR, EN and VU) in drylands (≥50% global range in drylands). Percentages show the proportion of species assessed affected by threats per taxon. Most species are subjected to multiple threats.

vertebrate groups found in drylands globally (total $n$ = 3,589)−80% have less than 10% of their dryland distributions protected (Fig. 3d, inner circles), 53% are endemic to drylands and 13% are listed as threatened by the IUCN Red List (categories Critically Endangered (CR), Endangered (EN) and Vulnerable (VU); Fig. 3d, outer circles). Australia is a hotspot of reptile richness in drylands, with 726 species, 58% of which are endemic to drylands; 75% of all species have less than 10% of their distributions protected, and 5% are threatened (Extended Data Fig. 4). Other regions also emerge as important for dryland biodiversity due to their species richness. For example, South America is a hotspot of dryland biodiversity across all vertebrate groups (Extended Data Fig. 4) especially considering its relatively small area (9% of global drylands; Table 1). African and Asian drylands combined, matching their vast scale, host the highest richness of vertebrate species−45% ($n$ = 3,627) of global dryland vertebrate species (Supplementary Data 1).

In addition, we evaluated the protection of dryland species by setting more demanding representation targets of protected area coverage for species with restricted ranges[44]. We set a 100% protection target for species with ranges smaller than 1,000 km² in drylands and a 10–100% protection target for species with intermediate and widespread ranges above 1,000 km². We find that most narrow-ranging species in drylands are inadequately protected (Fig. 4 and Supplementary Data 2). For example, 33% of amphibian species in drylands have very narrow ranges ($n$ = 331); of these only 24 species have adequate protection, while 193 species have zero protection. Similarly, 18% of reptiles have very narrow ranges ($n$ = 629); of these only 36 species have adequate protection, whereas 431 species have zero protection. Narrow-ranging birds ($n$ = 65) and mammals ($n$ = 96) are less common; however, of these only 2 birds and 3 mammals are adequately

protected. Such protection gaps are concerning, as narrow-ranging species are in greatest need of protection due to various threats and are consequently often listed as threatened by the IUCN (for example, 69% of narrow-ranging bird species are threatened compared to 4% of widespread species; Extended Data Fig. 5). Intermediate and widespread ranging species across taxa are slightly better protected according to representation targets in drylands (amphibians = 18%; birds = 18%; mammals = 16%; reptiles = 16%; Fig. 4); nevertheless, most dryland vertebrate species are inadequately protected.

We further explored threat types currently affecting dryland species as designated by the IUCN threat-classification scheme per species (v.3.3; Fig. 5). The largest and most notable anthropogenic threat to vertebrate groups in drylands is agriculture, as reported in other regions[45–47] (Extended Data Fig. 6). Other prevalent threats in drylands include timber and plant harvesting (for example, harvesting plants and trees for commercial, recreational and subsistence uses), threats from invasive species and disease, and infrastructure development. However, different threats emerge as important across different groups in drylands. For example, amphibians are most impacted by water management in drylands (20% of species assessed) compared to the other groups (Fig. 5a), climate change is a larger threat to birds (which are especially vulnerable to heatwaves)[29] in drylands (29% of species assessed; Fig. 5b), over-exploitation (that is, direct hunting and harvesting of animals) is a greater threat to mammals (42% of species assessed; Fig. 5c), and infrastructure development (that is, residential/commercial development and transportation/service corridors) and mining/energy production threaten reptiles relatively more than the other groups (37% and 21% of reptile species assessed, respectively; Fig. 5d). The IUCN has yet to comprehensively review dryland regions

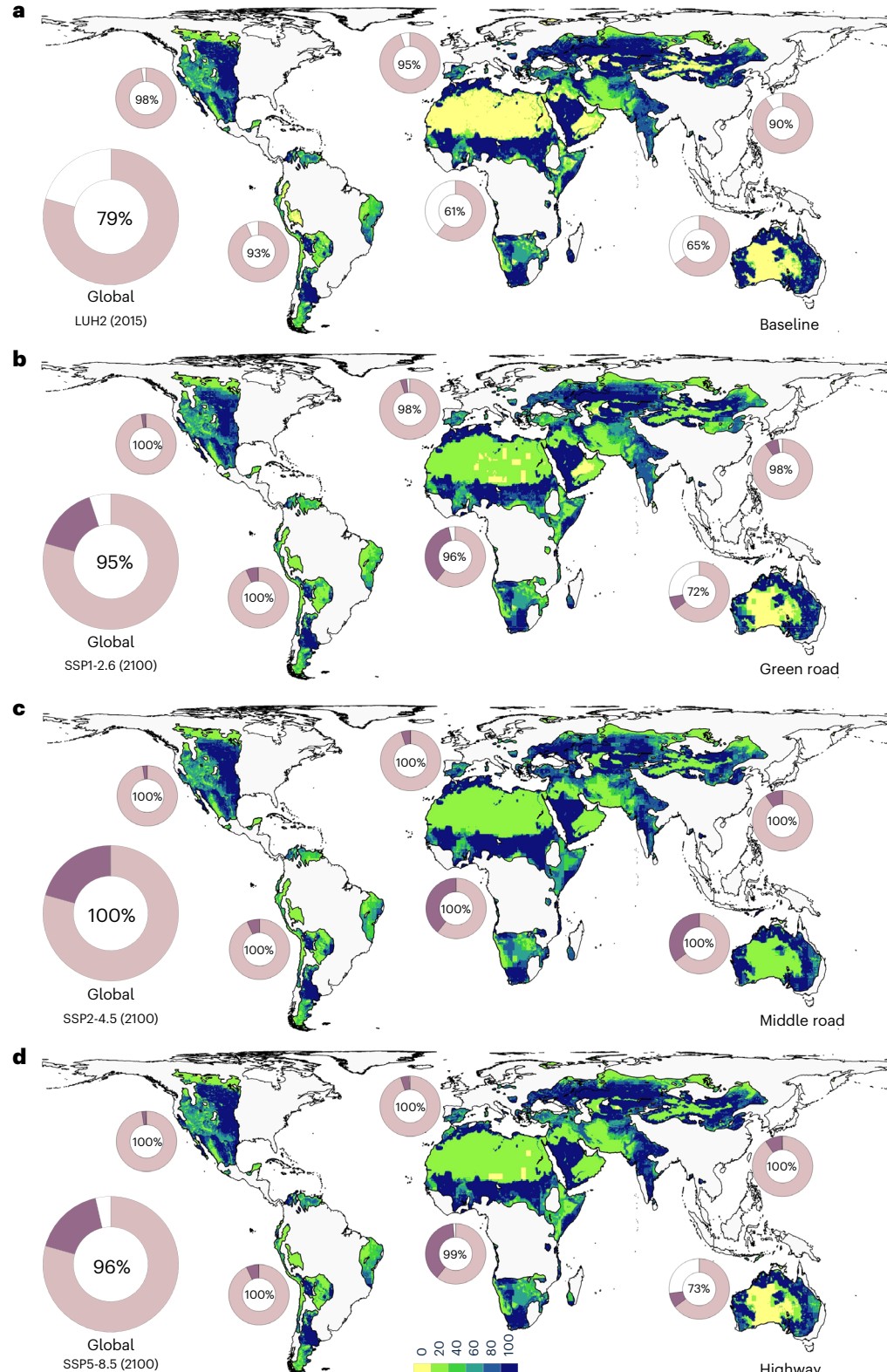

**Fig. 6 | Current and future land-use scenarios in drylands. a–d,** Current land-use patterns in drylands, LUH2 (2015) (**a**) and under SSPs: SSP1-2.6 (2100) (**b**), SSP2-4.5 (2100) (**c**) and SSP5-8.5 (2100) (**d**). Colour codes correspond to fraction of grid cell occupied by cropland, pasture, rangeland and urban land classes from low (yellow) to high (blue). Circles show total land cover area of 25 km² grids occupied by some fraction of land classes above 0% (that is, total fractional land areas). Pink, total fractional land areas in 2015 (**a**); purple, projected additional total fractional land areas (**b**–**d**).

and threats to dryland species, and therefore true assessments of threat are not fully known[11,48], with potentially important implications regarding our understanding of threats to dryland biodiversity. This is especially true for reptile species—38% in drylands are unassessed or designated as data deficient by the IUCN compared to 24% elsewhere. These knowledge gaps compound the threat to dryland fauna, as the

status of unassessed species or those designated as data deficient have been shown to be more similar to species identified as threatened by the IUCN[49,50]. Overall, most dryland vertebrate species including dryland endemic species and narrow-ranging species are inadequately protected and therefore require more attention in the context of global targets of biodiversity protection, especially due to the vulnerability of drylands to acute land-use threats and to projected future increasing human pressures.

## Current and future land-use change scenarios in drylands

Drylands are especially vulnerable to anthropogenic activities driving biodiversity loss[10,13,15,51,52]. Furthermore, human populations and rates of urbanization are projected to increase dramatically faster in drylands compared to other regions, especially in Africa and Asia[53–55]. We evaluated cropland, rangeland, pasture and urban land-use pressures in drylands currently and under future climate and global socioeconomic change scenarios (that is, shared socioeconomic pathways, SSPs)[56]. We calculated the total land cover area of 25 km$^2$ grids occupied by some fraction of the above land types (that is, fractional land-use patterns)[57] and additional fractional land areas occupied under future projected scenarios. We find that under different future scenarios by 2100, most drylands across continents are projected to be converted by some degree due to anthropogenic land uses (Fig. 6a–d). At current policy trends, moderate population growth and sustainable consumption of resources and energy (SSP2-4.5), 100% of drylands are projected to be converted to human land uses by some fraction (12.6 × 10$^6$ km$^2$ additional natural drylands will be converted; Fig. 6c). This pattern persists even under the most optimistic future scenario (SSP1-2.6), characterized by low emissions with moderate socioeconomic trends limiting global biodiversity loss and environmental impacts (that is, 'green growth')[57]. Again, most natural drylands are projected to be converted by some fraction under such a scenario (9.6 × 10$^6$ km$^2$ additional natural drylands will be converted; Fig. 6b), especially as a result of increases in cropland and urban areas with concomitant reductions in pasture and rangeland (including unmanaged grasslands and shrublands with native vegetation)[58]—a trend we find across different socioeconomic pathways (Extended Data Figs. 7–10). Therefore, large areas of natural drylands are projected to be converted as a result of increasing cropland and urban areas, while large areas of rangelands (including native shrublands, deserts and so on) are predicted to be converted as well. This is mostly a consequence of the vast land resources needed for agriculture[10] and alternative energy production (that is, solar panels and bioenergy)[59]. Moreover, the development of solar energy resources has wide-ranging impacts on dryland ecosystems and biodiversity through habitat fragmentation and loss[60]. Also, the substantial expansion of urban land types and infrastructure in drylands including artificial structures and surfaces, roads and mining/extraction sites[61,62] are expected to have extensive impacts on dryland species[63–65] (Fig. 5). Such threats are likely to have multiple and interactive effects on dryland species further exacerbated by climate change and habitat fragmentation[66]. Similarly, in a projected extreme scenario characterized by rapid and resource-intensive development and high emissions (SSP5-8.5), most natural drylands are projected to undergo some degree of conversion to human land uses (10.4 × 10$^6$ km$^2$ additional natural drylands will be converted; Fig. 6d). However, these conversion rates are slightly lower than the 'middle road scenario'[57] (SSP2-4.5; Fig. 6c), mostly due to extreme climate and water restrictions limiting cropland, pasture and rangeland expansion in some dryland regions (Extended Data Fig. 10), coupled with declines in global population[67]. Ultimately, drylands are projected to face considerable human land-use changes irrespective of projected socioeconomic scenarios. These trends greatly threaten the future integrity of dryland systems with important implications for biodiversity conservation, especially considering that the vast majority of dryland habitats and species are not adequately protected.

## Conclusions

We evaluated the global distribution of drylands to assess their protected area coverage, unique biodiversity and current and future threats—to highlight the urgent need for the strategic development of conservation targets in drylands. We found that dryland systems host high levels of unique species and habitat diversity that are grossly under-protected. Moreover, natural dryland habitats are perilously threatened by projected human population growth, land-use and climate changes. This is also true under very optimistic scenarios that aim to reduce global agricultural land coverage and promote progressive climate policies, which have dire consequences and other negative spillovers for drylands—for example, the increased conversion of land for alternative energy sources[52,68,69]. At the same time, drylands provide unique opportunities and considerable scope for achieving conservation and biodiversity goals, especially in Africa and Asia, including the expansion of protected area networks and incorporating stricter forms of protection, while recognizing community-based approaches to conservation[70,71] and other management strategies[47,72]. Greater emphasis on the inclusion of drylands within main frameworks for conservation and development priorities has the potential to notably contribute towards achieving global targets of biodiversity conservation while preserving valuable ecological and human systems in drylands[11,14,30].

## Methods
### Global drylands

We evaluated the geographic extent of drylands as classified by the UNEP-WCMC[31] using the 'Drylands Dataset (2007)'. Drylands are categorized based on aridity values defined as the ratio of precipitation to potential evapotranspiration ($P$/PET), with values below 0.65. Drylands are further distinguished between subtypes using aridity intervals—hyperarid ($P$/PET < 0.05), arid ($P$/PET = 0.05–0.20), semiarid ($P$/PET = 0.20–0.50) and dry subhumid ($P$/PET = 0.50–0.65). We excluded 'presumed drylands'—additional areas that do not fall within the above intervals (that is, $P$/PET ≥ 0.65) but may contain 'dryland features'—data downloaded from https://datadownload.unep-wcmc.org/datasets. Data were processed using ArcGIS (v10.8.1.), projected using an equal-area Behrmann projection and overlaid with world continents ('World Continents 2023'; Esri—data downloaded from http://hub.arcgis.com/datasets/esri::world-continents/), omitting Oceania and Antarctica due to their low proportion of dryland area. To determine the proportion of biomes overlapping with drylands (that is, biome area in drylands/global drylands area), we overlaid World Wide Fund for Nature biomes[1] with drylands (Extended Data Fig. 1).

### Protected area coverage

To assess the degree of protected area coverage of drylands (that is, proportion of protected dryland area), we overlaid drylands with the WDPA[32] (data downloaded on November 2023 from www.protectedplanet.net/). Some regions lacked updated data (that is, China), and therefore for China we merged WDPA data from 2016 (when data were last reported fully) with those from 2023. Other regions did not provide complete WDPA data (that is, Turkey and India), and therefore for Turkey and India we compiled data on protected areas using OpenStreetMap[33] (data downloaded on November 2023 from www.openstreetmap.org/; see Supplementary Fig. 1 for comparisons to WDPA 2023 data). We conducted this protected area analysis at two levels: (1) with protected areas defined by the IUCN management categories 'I–IV', which are well suited for protecting biological diversity and restricting human activities[37] and (2) for all categories of protected areas—using IUCN management categories of protected areas 'I–IV' and categories 'V and VI' (permitting resource extraction and mixed land

uses), including protected areas that lack an explicit IUCN management category for various reasons (that is, categorized as 'not applicable', 'not assigned' and 'not reported')—herein 'uncategorized'. We omitted protected areas without polygon boundaries. We cleaned the data using the wdpar[73] package (v.1.3.7) and the function 'wdpa_clean' in R v4.2.2. We excluded 'Other Effective Conservation Measures', defined as conservation areas other than a protected area[74], because most countries have not yet provided data to the WDPA on Other Effective Conservation Measures[47].

### Species distribution data, range protection, IUCN threats and representation targets

We obtained the extent of occurrence polygons for all breeding, extant and native amphibians and mammals from the IUCN (v.6.3; www.iucnredlist.org/), birds from BirdLife International (v.4; http://datazone.birdlife.org/) and reptiles from the Global Assessment of Reptile Distributions (GARD v.1.7)[12]. We defined land vertebrate species in drylands as those with ≥50% of their global distributions occurring in drylands (by overlaying species polygons with drylands) for a final dataset of 1,001 amphibians, 2,000 birds, 1,493 mammals and 3,589 reptiles inhabiting drylands. We defined species endemic to drylands as those with >99% of their global distributions occurring in drylands.

For each species, we identified its spatial overlap with protected areas in IUCN categories I–IV (see 'Protected area coverage'). We categorized species as those with the following: (1) 0% protection (that is, <0.1% range overlap with protected areas), (2) 0–10% protection (that is, ≥0.1% to <10% range overlap with protected areas) and (3) 10–100% protection (that is, ≥10% range overlap with protected areas). We conducted these analyses at the global scale (Fig. 3) and at the continental level (Extended Data Fig. 4 and Supplementary Data 1). We further evaluated the threat status of dryland species, highlighting those species identified as threatened by the IUCN Red List categories—VU, EN and CR—data obtained from www.iucnredlist.org. For 372 reptile species yet to be assessed by the IUCN (as of June 2022), we used modelled threat assessment categories from ref. 50. We repeated the above analyses for species with ≥90% of their global ranges in drylands and find similar trends as reported above for species with ≥50% of their distributions occurring in drylands (Supplementary Figs 2–4).

In addition, we evaluated the protection of dryland species by setting representation targets of protection for species according to range size (Fig. 4). We set a 100% protection target for species with ranges <1,000 km² in drylands and a 10–100% protection target for species with intermediate ranges (1,000–250,000 km²) and widespread ranges (>250,000 km²). We categorized range protection as follows: 0% protection (<0.1% of range size protected), 0–10% (0.1–9.9%), 10–100% protection (9.9–99%) and 100% protection (>99%). We repeated these analyses at the continental level (Supplementary Data 2).

We further assessed the specific types of threat affecting dryland species in IUCN threatened categories (VU, EN, CR; Fig. 5) using the IUCN threat-classification scheme v.3.3—providing comprehensive data on known threats per species (data obtained from www.iucnredlist.org). We grouped similar threats to allow for simpler comparison according to ref. 45, except for the grouping of 'invasive species', which we combined with diseases under the grouping 'invasion and disease' according to ref. 46. When relevant, multiple threats were coded per species (for example, 'agriculture' and 'over-exploitation'). Following ref. 46, we considered only future and ongoing threats, we omitted threats affecting only a minority of the global population (that is, <50% of the population) per species, and we removed 'negligible' threats and those causing 'no declines'.

### Current and future land-use change scenarios

To investigate current and future human land uses in drylands, we extracted data from the Land Use Harmonization (LUH2) project[57] (data obtained from http://luh.umd.edu/). The LUH2 dataset integrates historical human land-use data, management activities and maps with projected models simulating future land-use changes under different scenarios. These models comprise a number of sub-models describing agricultural, demographic, socioeconomic, vegetation and climate systems operating at several different spatial resolutions (for example, local, regional and national)[59]. Therefore, the harmonization strategy estimates fractional land-use patterns at a globally consistent scale of 0.25° grids integrating historical reconstructions with future projections of land use incorporating multiple sub-datasets. Recently released datasets at higher resolutions—see ref. 75 (for example, their Fig. 10), ref. 76 (for example, their Fig. 4), and ref. 77)—correlate strongly with LUH2 spatially for comparable land classifications; however, these datasets may have several limitations based on data availability yielding potential errors in future land projections. Also, these datasets do not distinguish between grazing areas (that is, rangeland, grassland and pasture regions), which have particular relevance and distinctions in drylands. We compared the LUH2 dataset applied here to drylands with a recently published dataset—the Future Land Use Simulation model[76]. When comparing global values of total land areas (10⁶ km) of similar land types under projected future scenarios, we find that the general trajectories are similar to LUH2 in drylands (Supplementary Fig. 5). Therefore, we use LUH2—widely applied in spatially explicit global land-use analyses (for example, refs. 78–80). However, these data contain several limitations including: their coarse scale, which may cause some deviations in land cover patterns and prevent local analyses; regional differences in estimating future land-use changes based on historical data availability; and additional uncertainties on biodiversity impacts regarding species-specific responses and sensitivities to different land types based on behavioural and physiological traits[28,81].

We focused on cropland, rangeland, pasture and urban land-use states, excluding natural vegetation (that is, primary or secondary forest or non-forest), to combine the following raster layers at 25 km² resolution: 'managed pasture', 'rangeland', 'urban land' and 'cropland'—comprising 'C3 annual crops', 'C3 perennial crops', 'C4 annual crops', 'C4 perennial crops' and 'C3 nitrogen-fixing crops'. For current conditions, we used LUH2 (v2h) from the year 2015 (providing single land-use estimates per grid without uncertainty intervals). To analyse future trends of land use in drylands, we used Phase 6 of the Coupled Model Intercomparison Project (CMIP6) from LUH2, which provides future scenarios based on alternative climate change and socioeconomic scenarios—SSPs[57]. We evaluated scenarios SSP1-2.6, SSP2-4.5 and SSP5-8.5. SSP1-2.6 represents a sustainability and low emissions scenario with mean global warming projected at below 1.8 °C by 2100 compared to pre-industrial levels (that is, 'green growth scenario'). SSP2-4.5 represents a moderate path scenario that does not deviate markedly from current patterns with warming expected at approximately 2.7 °C (that is, 'middle road'). SSP5-8.5 represents a high-resource and energy-intensive scenario with warming projected to be between 3.3 °C and 5.7 °C (that is, 'highway').

We overlaid land cover layers with drylands and identified the land-use fractions of grid cells of exactly 0.22180° resolution (projected using an equal-area Behrmann projection). Grid cells were determined as dryland if grid cell centres fall within the dryland polygon layer. We classified grids according to the following intervals to help with interpretation: (1) 0%, (2) 0–20%, (3) 20–40%, (4) 40–60%, (5) 60–80% and (6) 80–100%. We treated each land class equally without applying different weights[54] because impacts are challenging to weigh globally due to non-stationarity and regional variations and sensitivities of different animals and plants to different pressures. To calculate projected future land area of natural drylands converted, we summed the total land cover area of grids occupied by some fraction of the above land classes (that is, all grids above 0% land-use fraction)—herein 'total fractional land areas' plus the additional total fractional land areas occupied under future projected SSPs (that is, circles in

Fig. 6). Rangelands comprise vast areas of natural and unmanaged habitats and ecosystems including grasslands, shrublands, woodlands and deserts containing native vegetation and typically have low livestocking densities[36,58]—possibly exerting a disproportionate influence on the land-use patterns reported. Therefore, to quantify the relative contribution of rangeland to total fractional land areas, we repeated the above analysis for cropland, pasture and urban land classes combined excluding rangelands (Extended Data Fig. 7). We also analysed trends in the gain (increase in fraction of each grid cell occupied) or loss (decrease in the fraction of each grid cell occupied) for each land class under different SSPs relative to the baseline (land-use data from LUH2 in 2015) using zonal statistics in ArcGIS (Extended Data Figs. 8–10).

### Reporting summary

Further information on research design is available in the Nature Portfolio Reporting Summary linked to this article.

### Data availability

The Drylands Dataset (2007) was obtained from UNEP-WCMC (https://datadownload.unep-wcmc.org/datasets). Protected area coverage data were obtained from the World Database on Protected Areas (www.protectedplanet.net/) and OpenStreetMap (at www.openstreetmap.org/). Species distribution data are available for amphibians and mammals at the IUCN (v.6.3; www.iucnredlist.org/), birds at BirdLife International (v.4; http://datazone.birdlife.org/) and reptiles at the Global Assessment of Reptile Distributions (GARD v.1.7). IUCN Red List data and IUCN threat classifications were obtained from the IUCN (www.iucnredlist.org). Silhoutte images of vertebrate taxa were obtained without changes from PhyloPic (https://www.phylopic.org/)[82]. Land-use data were obtained from the LUH2 project (v2h) from the year 2015 (http://luh.umd.edu/). Future land-use data were obtained from phase 6 of the Coupled Model Intercomparison Project (CMIP6) from LUH2 (http://luh.umd.edu/). Source data are provided with this paper.

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

## Acknowledgements

This work was supported by the Daniel E. Koshland Interdisciplinary Research Fund (to S.R. and U.R.); it was also partially supported by the Israel Science Foundation number 611/23 (to U.R.) and partially supported by the Israeli Ministry of Science and Technology as part of the DesertData – The DeserTech Knowledge Center for Sustainability. We would like to thank Ernest Frimpong Asamoah for sharing 2016 WDPA data for China.

## Author contributions

Conceptualization: A.L., S.R. and U.R. Methodology: A.L., G.M. and U.R. Investigation: A.L., G.M. and U.R. Visualization: A.L., G.M. and U.R. Funding acquisition: S.R. and U.R. Supervision: S.R. and U.R. Writing—original draft: A.L. Writing—review and editing: A.L., G.M., S.R. and U.R.

## Competing interests

The authors declare no competing interests.

## Additional information

**Extended data** is available for this paper at https://doi.org/10.1038/s41559-024-02450-4.

**Correspondence and requests for materials** should be addressed to Amir Lewin.

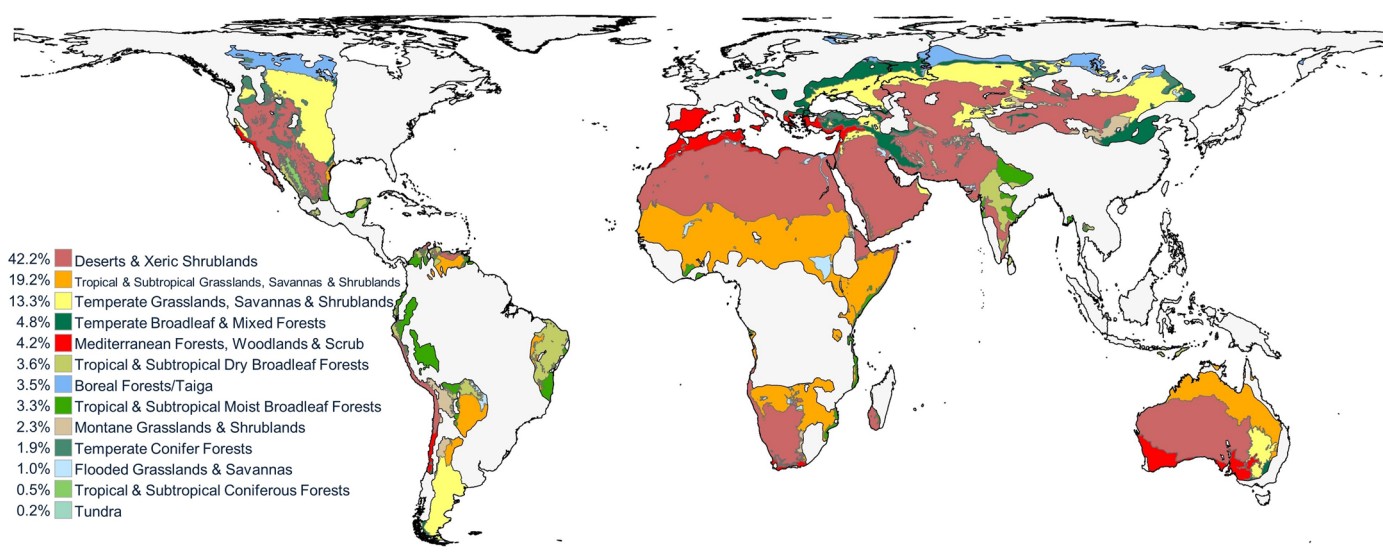

**Extended Data Fig. 1** | Spatial distribution and percent area of WWF biomes in drylands (that is, biome area in drylands/global drylands area).

Legend:
- 42.2% Deserts & Xeric Shrublands
- 19.2% Tropical & Subtropical Grasslands, Savannas & Shrublands
- 13.3% Temperate Grasslands, Savannas & Shrublands
- 4.8% Temperate Broadleaf & Mixed Forests
- 4.2% Mediterranean Forests, Woodlands & Scrub
- 3.6% Tropical & Subtropical Dry Broadleaf Forests
- 3.5% Boreal Forests/Taiga
- 3.3% Tropical & Subtropical Moist Broadleaf Forests
- 2.3% Montane Grasslands & Shrublands
- 1.9% Temperate Conifer Forests
- 1.0% Flooded Grasslands & Savannas
- 0.5% Tropical & Subtropical Coniferous Forests
- 0.2% Tundra

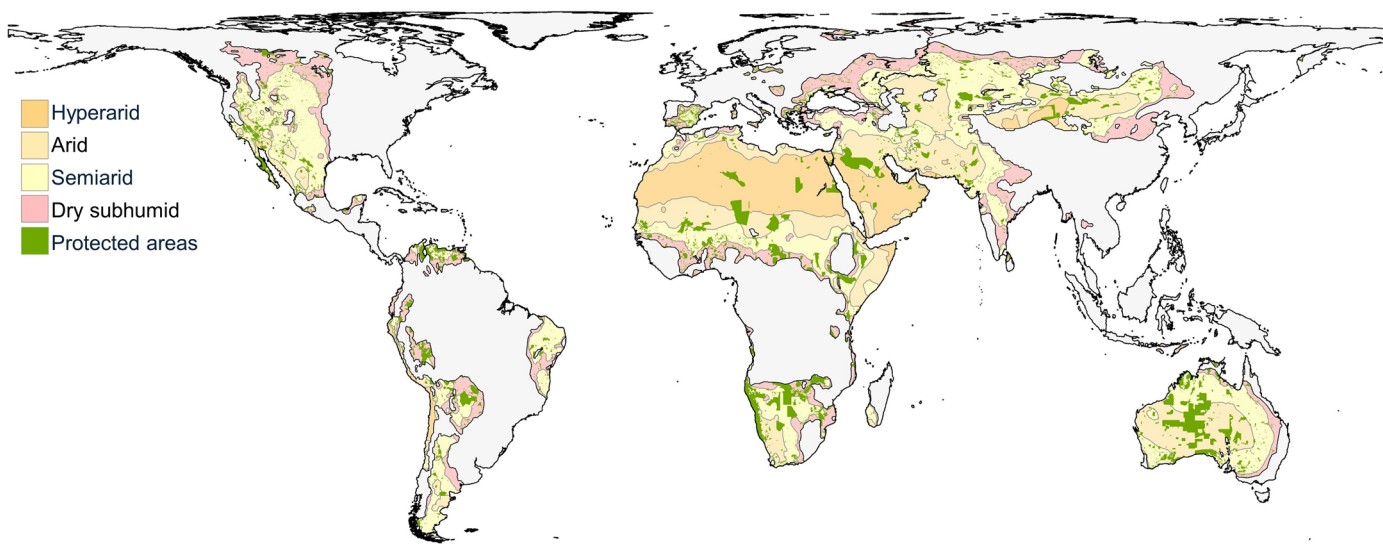

**Extended Data Fig. 2 | Protected areas of global drylands and subtypes.** Green = IUCN categories I-VI and uncategorized (that is, 'not applicable', 'not assigned' and 'not reported').

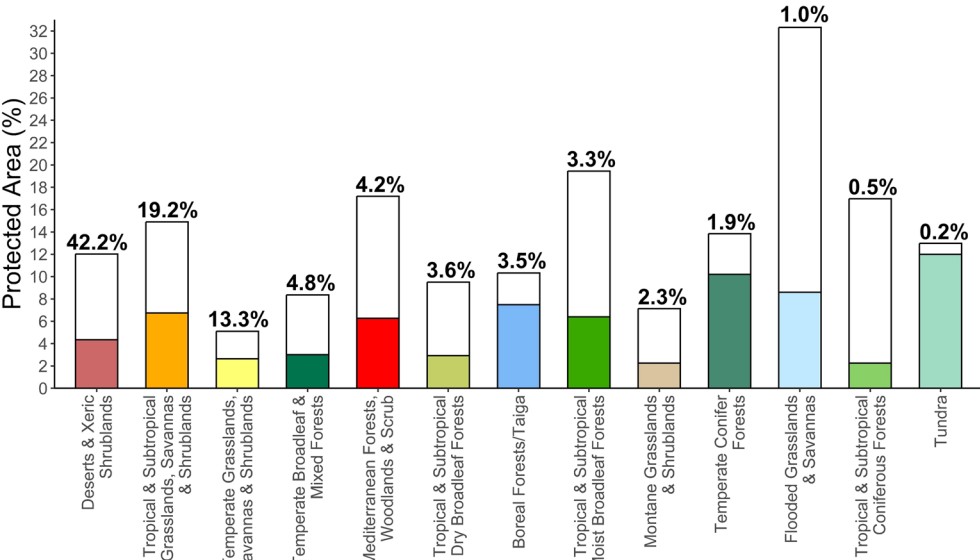

**Extended Data Fig. 3 | Protected areas of WWF biomes in drylands.** Coloured bars = IUCN categories I-IV; white bars = IUCN categories V-VI and uncategorized (that is, 'not applicable', 'not assigned' and 'not reported'). Biomes are in order of decreasing proportion of global drylands shown in percentage values (that is, biome area in drylands/global drylands area).

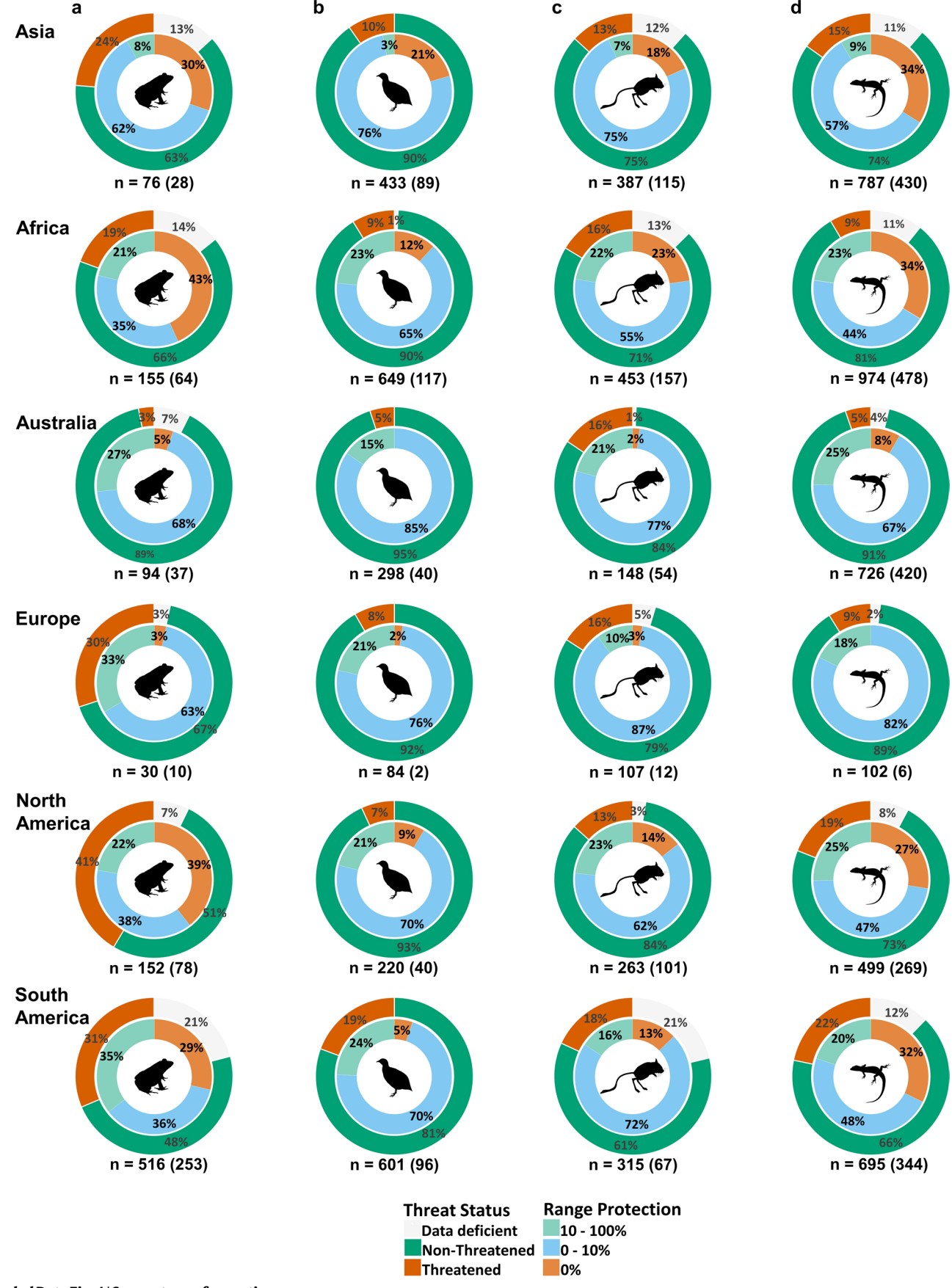

**Extended Data Fig. 4 | See next page for caption.**

**Extended Data Fig. 4 | Proportion range size protected and IUCN threat status of amphibian (a), bird (b), mammal (c), and reptile (d) species by continent in drylands (≥50% global range in drylands).** Inner circles show proportion of species with range size protected (IUCN categories I-IV): orange = 0% range protected, blue = 0-10% range protected, green = 10-100% range protected. Outer circles show proportion of IUCN threatened species: dark orange = threatened (IUCN categories CR, EN, VU), dark green = non-threatened (IUCN categories NT, LC), grey = data deficient (IUCN category DD). n = number of species, parentheses = number of endemic species (>99% range in drylands).

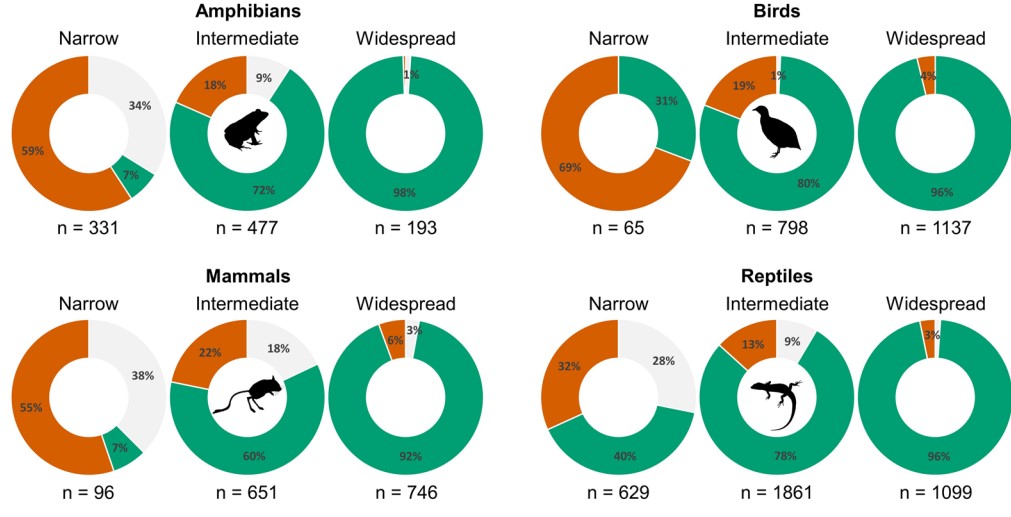

**Extended Data Fig. 5 | IUCN threat status of amphibian, bird, mammal, and reptile species in drylands (≥50% global range in drylands) by range size according to representation targets.** Narrow-ranging species = <1000 km²; intermediate = 1000-250,000 km²; widespread = >250,000 km². Circles show proportion of IUCN threatened species: dark orange = threatened (IUCN categories CR, EN, VU), dark green = non-threatened (IUCN categories NT, LC), grey = data deficient (IUCN category DD). n = number of species.

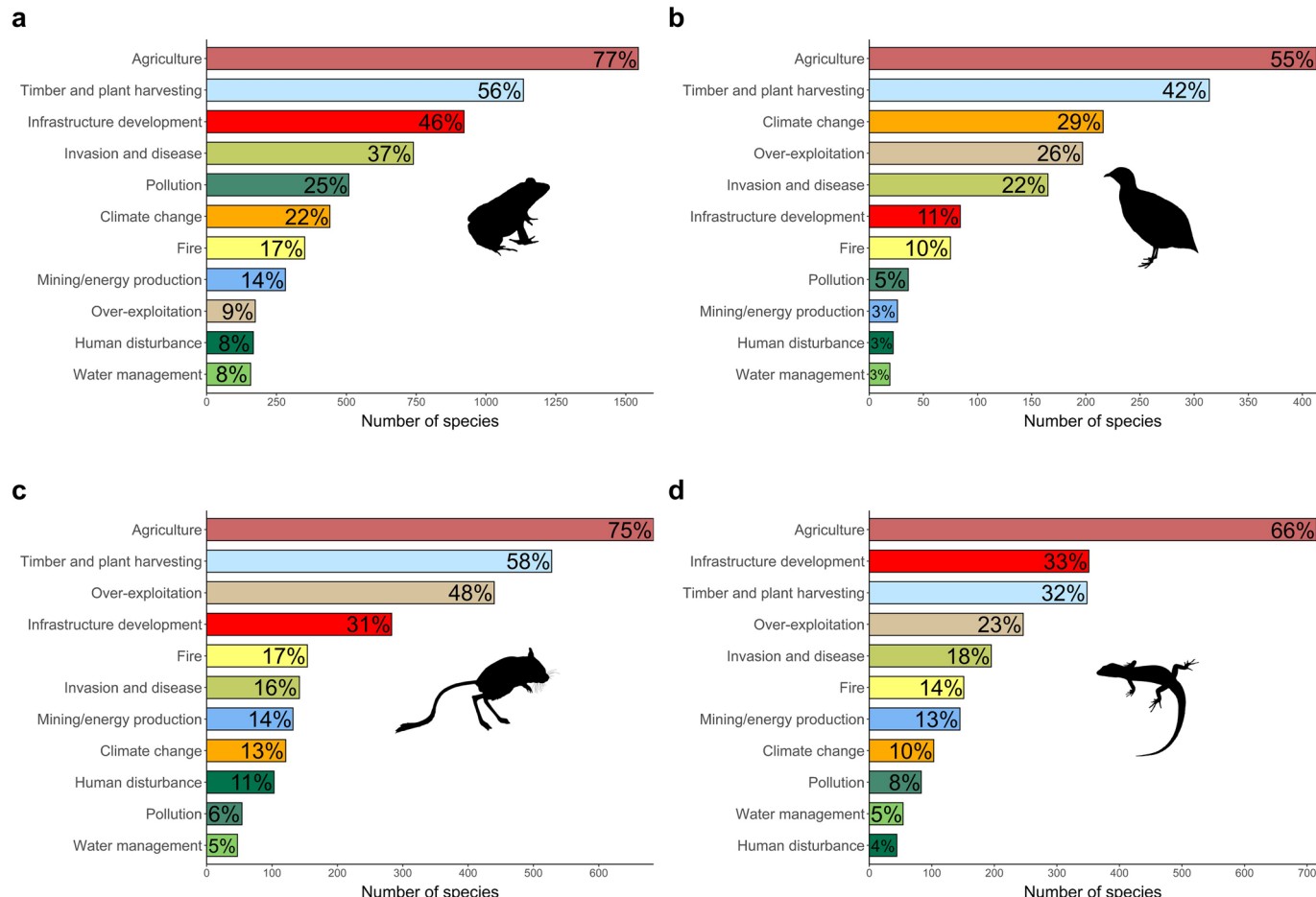

**Extended Data Fig. 6 | Types of threats affecting amphibian (a), bird (b), mammal (c), and reptile (d) threatened species (in IUCN categories CR, EN, VU) in non-drylands (≥50% global range in non-drylands).** Percentages show the proportion of species assessed affected by threats per taxon. Most species are subjected to multiple threats.

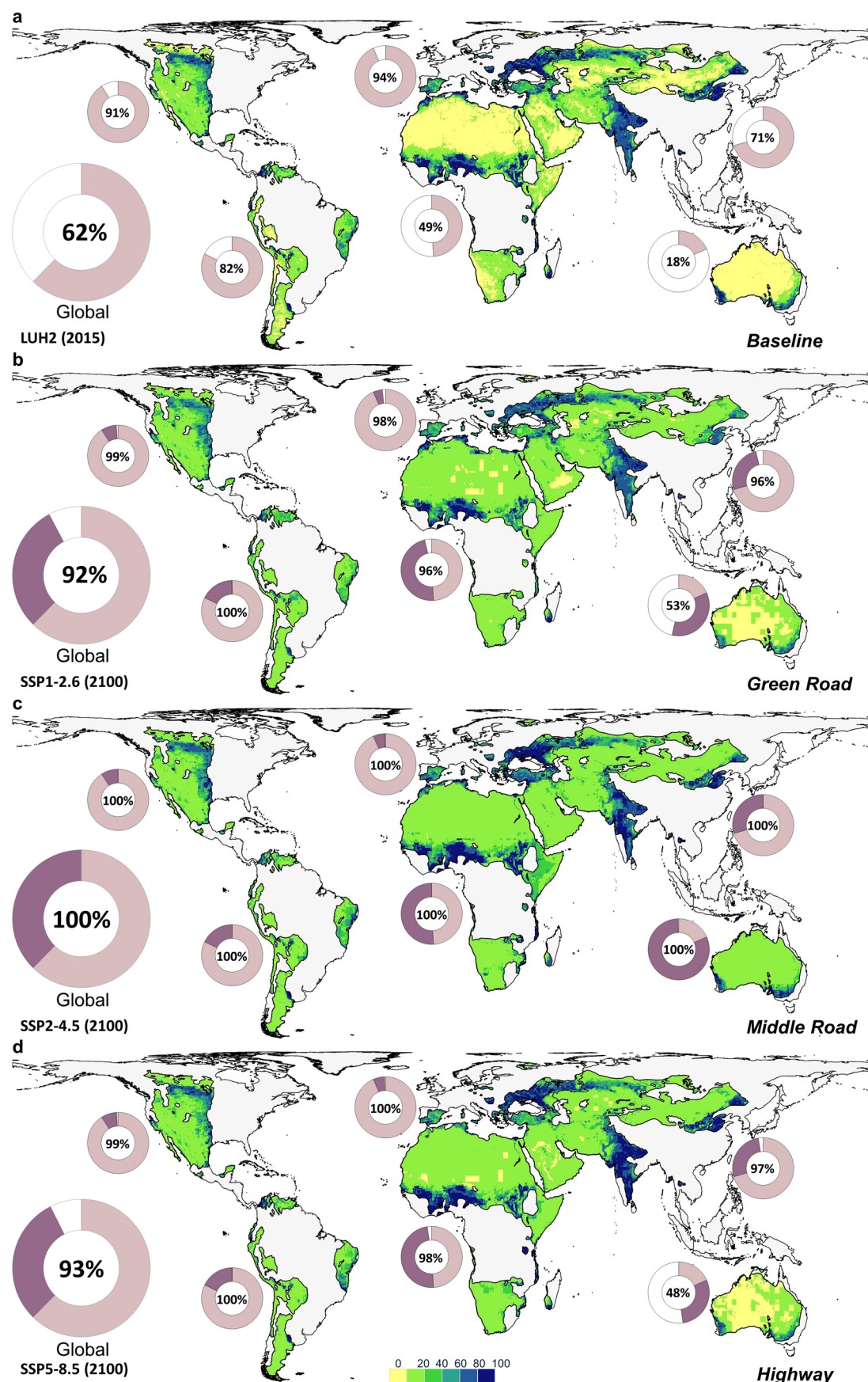

**Extended Data Fig. 7 | Current land-use patterns in drylands LUH2 (2015) (a), and under shared socioeconomic pathways (SSPs), SSP1-2.6 (2100) (b), SSP2-4.5 (2100) (c), and SSP5-8.5 (2100) (d).** Colour codes correspond to fraction of grid cell occupied by cropland, pasture, and urban land classes excluding rangeland from low (yellow) to high (blue). Circles show total land cover area of 25 km² grids occupied by some fraction of land classes above 0% (that is, total fractional land areas). Pink = total fractional land areas in 2015 (a), purple = projected additional total fractional land areas (b-d).

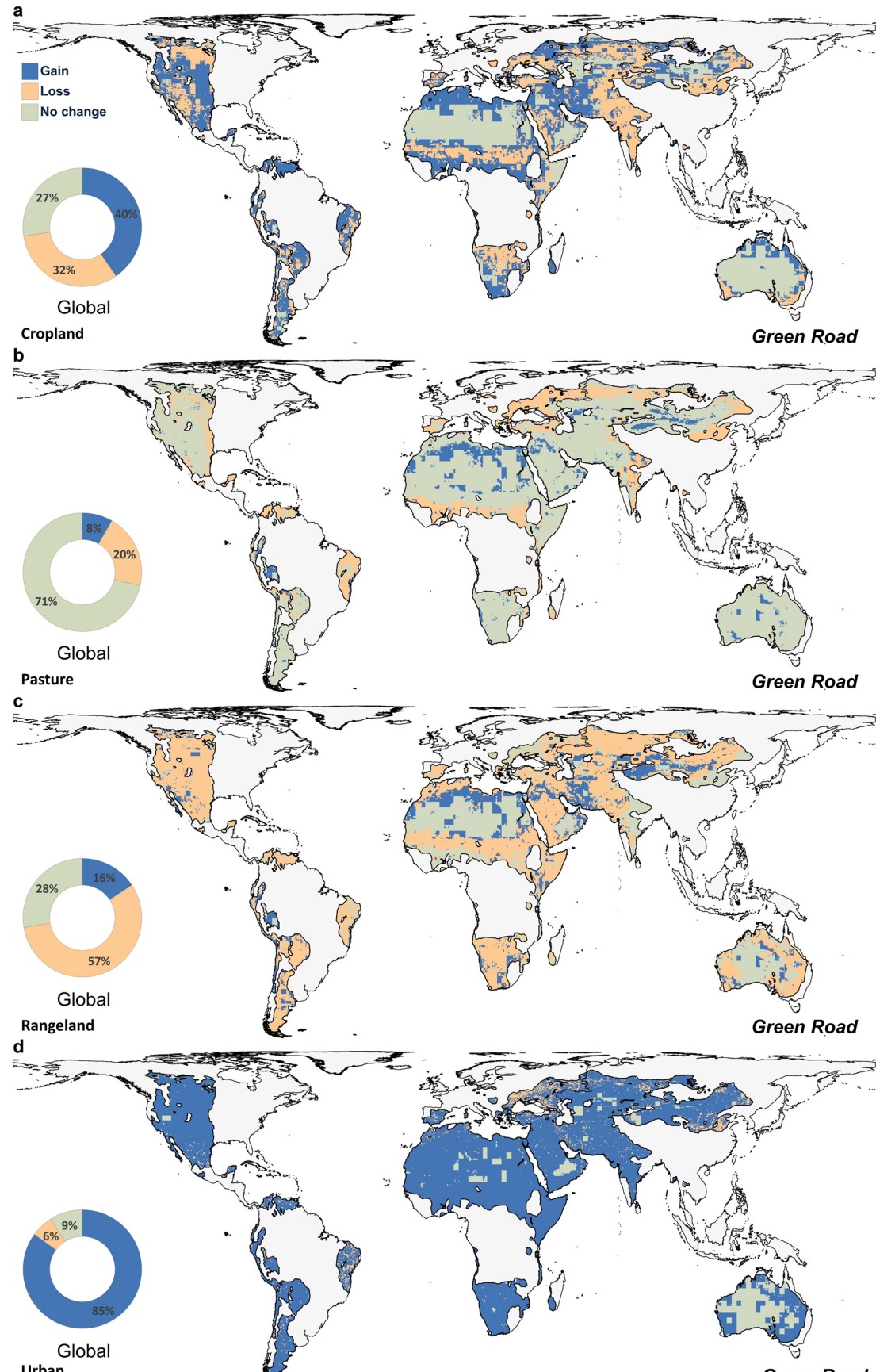

**Extended Data Fig. 8 | Change of land cover fraction of grids occupied by land types: cropland (a), pasture (b), rangeland (c), and urban land (d) relative to current values LUH2 (2015) under shared socioeconomic pathway (SSP), SSP1-2.6 (2100).** Blue = gain in land cover fraction, tan = loss, light green = no change.

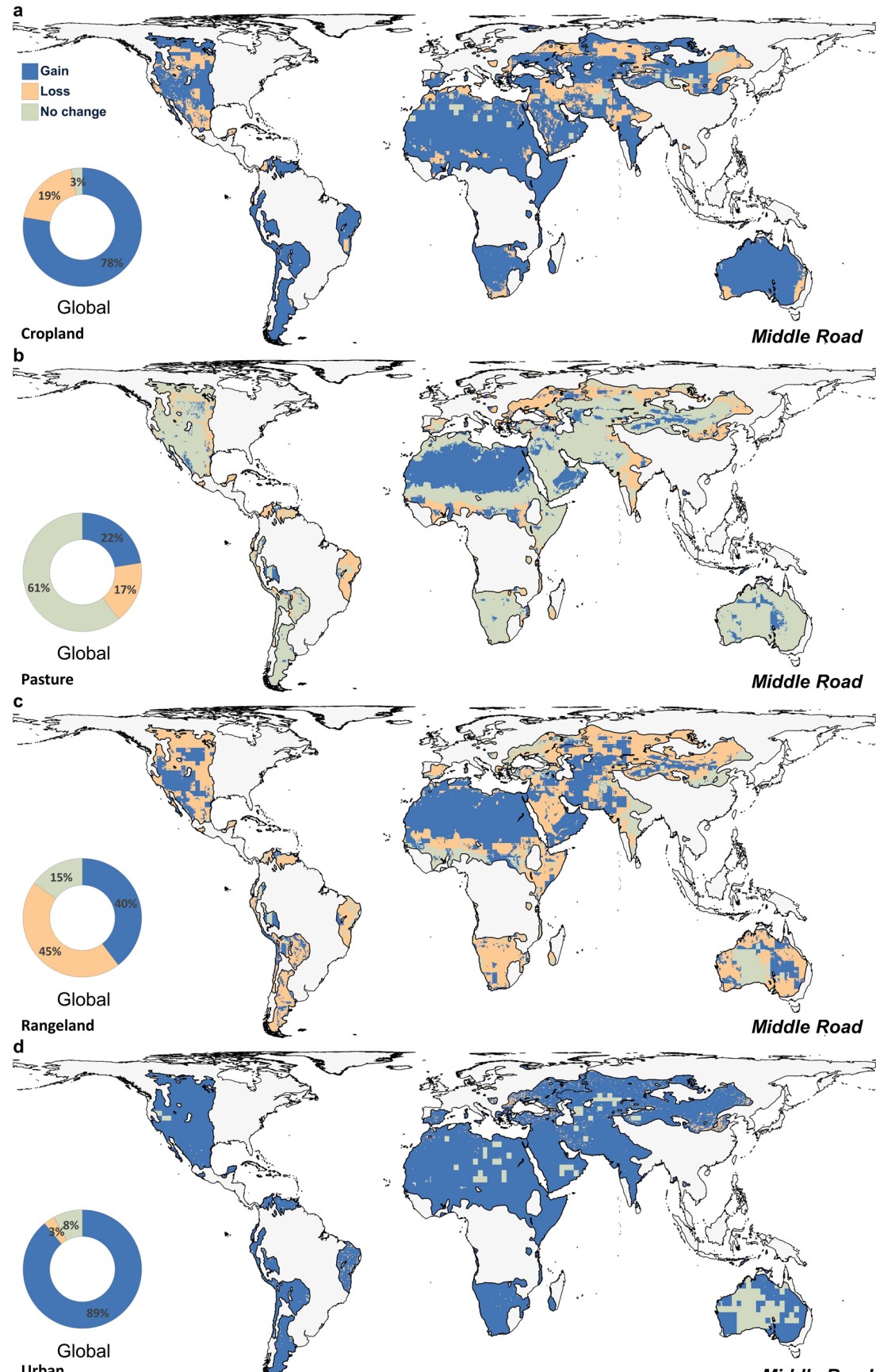

**Extended Data Fig. 9 | Change of land cover fraction of grids occupied by land types: cropland (a), pasture (b), rangeland (c), and urban land (d) relative to current values LUH2 (2015) under shared socioeconomic pathway, SSP2-4.5 (2100).** Blue = gain in land cover fraction, tan = loss, light green = no change.

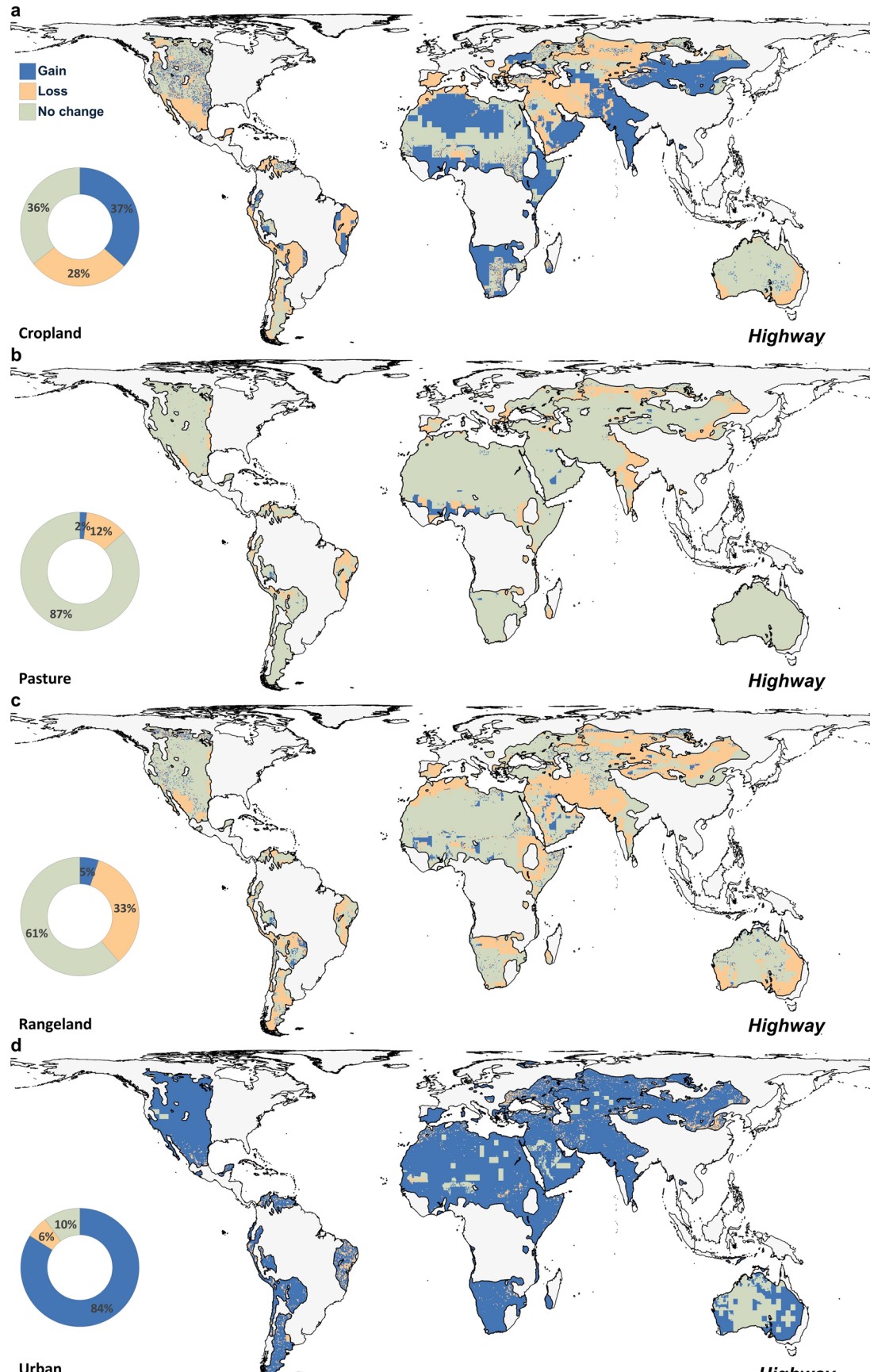

**Extended Data Fig. 10 | Change of land cover fraction of grids occupied by land types: cropland (a), pasture (b), rangeland (c), and urban land (d) relative to current values LUH2 (2015) under shared socioeconomic pathway, SSP5-8.5 (2100).** Blue = gain in land cover fraction, tan = loss, light green = no change.

# Reporting Summary

## Statistics

For all statistical analyses, confirm that the following items are present in the figure legend, table legend, main text, or Methods section.

| n/a | Confirmed | |
|---|---|---|
| ☒ | ☐ | The exact sample size (*n*) for each experimental group/condition, given as a discrete number and unit of measurement |
| ☒ | ☐ | A statement on whether measurements were taken from distinct samples or whether the same sample was measured repeatedly |
| ☒ | ☐ | The statistical test(s) used AND whether they are one- or two-sided<br>*Only common tests should be described solely by name; describe more complex techniques in the Methods section.* |
| ☒ | ☐ | A description of all covariates tested |
| ☒ | ☐ | A description of any assumptions or corrections, such as tests of normality and adjustment for multiple comparisons |
| ☒ | ☐ | A full description of the statistical parameters including central tendency (e.g. means) or other basic estimates (e.g. regression coefficient) AND variation (e.g. standard deviation) or associated estimates of uncertainty (e.g. confidence intervals) |
| ☒ | ☐ | For null hypothesis testing, the test statistic (e.g. *F*, *t*, *r*) with confidence intervals, effect sizes, degrees of freedom and *P* value noted<br>*Give P values as exact values whenever suitable.* |
| ☒ | ☐ | For Bayesian analysis, information on the choice of priors and Markov chain Monte Carlo settings |
| ☒ | ☐ | For hierarchical and complex designs, identification of the appropriate level for tests and full reporting of outcomes |
| ☒ | ☐ | Estimates of effect sizes (e.g. Cohen's *d*, Pearson's *r*), indicating how they were calculated |

*Our web collection on statistics for biologists contains articles on many of the points above.*

## Software and code

Policy information about availability of computer code

| Data collection | No software was used for data collection. |
|---|---|
| Data analysis | Spatial analyses were performed in ArcGIS (v10.8.1.)<br>Data analysis was performed in R version 4.2.2 using the following packages:<br>wdpar package and the function 'wdpa_clean' |

For manuscripts utilizing custom algorithms or software that are central to the research but not yet described in published literature, software must be made available to editors and reviewers. We strongly encourage code deposition in a community repository (e.g. GitHub). See the Nature Portfolio guidelines for submitting code & software for further information.

## Data

Policy information about availability of data

All manuscripts must include a data availability statement. This statement should provide the following information, where applicable:
- Accession codes, unique identifiers, or web links for publicly available datasets
- A description of any restrictions on data availability
- For clinical datasets or third party data, please ensure that the statement adheres to our policy

The Drylands Datatset (2007) was obtained from UNEP-WCMC (https://datadownload.unep-wcmc.org/datasets).
Protected area coverage data were obtained from the World Database on Protected Areas (www.protectedplanet.net/), and OpenStreetMap (from

www.openstreetmap.org/).
Species distribution data are available for amphibians and mammals from the IUCN (v.6.3; www.iucnredlist.org/), birds from BirdLife International (v.4; http://datazone.birdlife.org/), and reptiles from the Global Assessment of Reptile Distributions (GARD v.1.7).
IUCN Red List data and IUCN threat-classifications were obtained from the IUCN (www.iucnredlist.org).
Land use data was obtained from the Land Use Harmonization (LUH2) project (v2h) from the year 2015 (http://luh.umd.edu). Future land use data was obtained from phase 6 of the Coupled Model Intercomparison Project (CMIP6) from LUH2 (http://luh.umd.edu/).

## Research involving human participants, their data, or biological material

Policy information about studies with [human participants or human data](). See also policy information about [sex, gender (identity/presentation), and sexual orientation]() and [race, ethnicity and racism]().

| | |
|---|---|
| Reporting on sex and gender | Not applicable |
| Reporting on race, ethnicity, or other socially relevant groupings | Not applicable |
| Population characteristics | Not applicable |
| Recruitment | Not applicable |
| Ethics oversight | Not applicable |

Note that full information on the approval of the study protocol must also be provided in the manuscript.

# Field-specific reporting

Please select the one below that is the best fit for your research. If you are not sure, read the appropriate sections before making your selection.

☐ Life sciences      ☐ Behavioural & social sciences      ☒ Ecological, evolutionary & environmental sciences

For a reference copy of the document with all sections, see nature.com/documents/nr-reporting-summary-flat.pdf

# Ecological, evolutionary & environmental sciences study design

All studies must disclose on these points even when the disclosure is negative.

| | |
|---|---|
| Study description | A global assessment of the degree of protected area and threats to biodiversity in drylands due to current and future human-induced land-use changes |
| Research sample | See Data description above |
| Sampling strategy | All relevant data were used. No statistical methods were used to predetermine sample size |
| Data collection | Data were based on existing datasets and was collected online by the authors |
| Timing and spatial scale | Global data from 2015 to 2100 |
| Data exclusions | No data were excluded from the analyses |
| Reproducibility | All data are accessible online |
| Randomization | Not applicable |
| Blinding | Not applicable |

Did the study involve field work?      ☐ Yes      ☒ No

# Reporting for specific materials, systems and methods

We require information from authors about some types of materials, experimental systems and methods used in many studies. Here, indicate whether each material, system or method listed is relevant to your study. If you are not sure if a list item applies to your research, read the appropriate section before selecting a response.

## Materials & experimental systems

| n/a | Involved in the study |
|-----|----------------------|
| ☒ ☐ | Antibodies |
| ☒ ☐ | Eukaryotic cell lines |
| ☒ ☐ | Palaeontology and archaeology |
| ☒ ☐ | Animals and other organisms |
| ☒ ☐ | Clinical data |
| ☒ ☐ | Dual use research of concern |
| ☒ ☐ | Plants |

## Methods

| n/a | Involved in the study |
|-----|----------------------|
| ☒ ☐ | ChIP-seq |
| ☒ ☐ | Flow cytometry |
| ☒ ☐ | MRI-based neuroimaging |

## Plants

| Seed stocks | Not applicable |
|-------------|----------------|
| Novel plant genotypes | Not applicable |
| Authentication | Not applicable |

