## [Peer Review File · Nature Ecology & Evolution]

Peer Review Information

Journal: Nature Ecology & Evolution

Manuscript Title: Global evaluation of current and future threats to drylands and their vertebrate biodiversity

Corresponding author name(s): Amir Lewin

Editorial Notes:

Redactions – published data

Reviewer Comments & Decisions:

Decision Letter, initial version:

31st October 2023

Dear Dr Lewin,

Your manuscript entitled "Current and future threats to global drylands" has now been seen by 3 reviewers, whose comments are attached. The reviewers have raised a number of concerns which will need to be addressed before we can offer publication in Nature Ecology & Evolution. We will therefore need to see your responses to the criticisms raised and to some editorial concerns, along with a revised manuscript, before we can reach a final decision regarding publication.

Firstly, like Referee #3, The Nature Ecology and Evolution Editorial team would like to express our profound sympathies with you in light of the ongoing situation in the Middle East. We recognise that you and your coauthors will have other immediate priorities in your personal lives, and we would like to confirm that we are not expecting any revisions to your manuscript to be submitted under any deadline from us. Please feel free to prepare your revised files whenever you are ready.

As you can see from the reports attached, all three referees are generally positive about the work, but felt that the first section on dryland designations was a little disconnected from the rest of the analysis. As such, they recommend a greater focus on the other sections, moving the first section to a supplementary file or summary Table. Referee #1 also feels that some further validation is required, while Referee #3 thinks that some more thorough assessment of systematic data gaps is needed for the protected area analysis.

We therefore invite you to revise your manuscript taking into account all reviewer and editor comments. Please highlight all changes in the manuscript text file [OPTIONAL: in Microsoft Word format].

* Include a "Response to reviewers" document detailing, point-by-point, how you addressed each

2reviewer comment. If no action was taken to address a point, you must provide a compelling argument. This response will be sent back to the reviewers along with the revised manuscript.

* If you have not done so already please begin to revise your manuscript so that it conforms to our Article format instructions at <http://www.nature.com/natecolevol/info/final-submission>. Refer also to any guidelines provided in this letter.

[REDACTED]

Nature Ecology & Evolution is committed to improving transparency in authorship. As part of our efforts in this direction, we are now requesting that all authors identified as 'corresponding author' on published papers create and link their Open Researcher and Contributor Identifier (ORCID) with their account on the Manuscript Tracking System (MTS), prior to acceptance. ORCID helps the scientific community achieve unambiguous attribution of all scholarly contributions. You can create and link your ORCID from the home page of the MTS by clicking on 'Modify my Springer Nature account'. For more information please visit www.springernature.com/orcid.

[REDACTED]

Reviewer expertise:

Reviewer #1: Drylands

Reviewer #2: Drylands

Reviewer #3: Protected Areas

Reviewers' comments:

Reviewer #1 (Remarks to the Author):

In this manuscript, the authors "evaluated four commonly used global delineations of drylands to assess their overlap, the formal degree of protection of global drylands, their land-vertebrate biodiversity, and projected human-induced threats to drylands under different future climate change and socioeconomic scenarios". This work deals with a very interesting topic, which is undoubtedly of

interest to Nature Ecology and Evolution readers. The paper was also very well written and was an enjoyable read. I value the attempt made by the authors and I found the results potentially far-reaching. However, at the same time I am skeptical with some of the take-home messages given by the authors because of the limitations of the methodological approach they used (see specific comments below). I will comment below on the three main sections of the paper.

Regarding the Dryland delineations section, my feeling is that this section is somewhat disconnected from the rest and it is not fully clear the value it adds to the manuscript. On the one hand I have the feeling that the authors are somehow artificially trying to make the case for disagreements and debates that I feel that are not dominant when they include sentences such as "A major obstacle for conservation planning in drylands is the lack of a consistent definition of what constitutes drylands⁹", "There is much overlap between these delineations, without a widely accepted definition to categorize drylands³². This lack of clarity can greatly hamper broad-scale conservation efforts in drylands³³" or "Such incongruencies among dryland delineations may have important implications for guiding scientific discourse³⁵, land management, and identifying broad-scale dryland conservation priorities³⁶." To me the UNEP definition is widely used and accepted, both within the scientific and policy communities, and thus is not as debated as suggested. Indeed, it is not surprising to find that is the definition used by the authors in the analyses they present in the text. But most importantly, my feeling is that in this exercise, while interesting, the authors are trying to compare classifications that are not fully comparable. Of course, one can expect to find large differences if we compare a classification based purely on climatic variables (such as the UNEP one) with other classification based on biomes or agricultural potential, such as the WWF or FAO ones. Different classifications have different objectives and I was not convinced that the analyses presented in the first part of the paper, while undoubtedly interesting, add anything relevant to the paper. Furthermore, if the incongruencies between the classifications mentioned in sentences such as those highlighted above are really problematic for delineating conservation/management actions then this should be reflected in the results of the analyses done by the authors, right? But this does not seem to be the case as reported by the authors ("Parallel analyses for the other delineations are provided in Extended Data and Supplementary Table S1, revealing similar trends as described below."). Thus, if the authors want to retain this part of the manuscript, I think they should provide additional justification of the value, soundness and need of the comparison among classifications they did.

I really liked the second part, Protected areas and biodiversity threats across drylands, which presents very interesting and relevant findings in a very clear manner. Nothing to object to this part.

I am, however, particularly concerned about the analyses and messages given in the third part of the manuscript, Current and projected anthropogenic threats to drylands. First, I have to say that I am not sure to have fully understood the analyses they did because the methodological part of the manuscript is very brief and do not provide all the explanations/details needed to allow readers to fully understand what they did. Regardless this important point, I am particularly worried with the land use intensity classification made by the authors, which is not properly explained/justified and indeed deliver results that are difficult to trust. If I properly understood what the authors did, they quantified the cover of the following land uses: 'managed pasture', 'rangeland', 'urban land', 'C3 annual crops', 'C3 perennial crops', 'C4 annual crops', 'C4 perennial crops' and 'C3 nitrogen-fixing crops' and then "overlaid all land-cover layers with drylands and identified the proportional cover of land cover types per pixel, which we then classified to the following land-use intensity categories to help with interpretation: (1) 0%, (2) 0-20%, (3) 20-40%, (4) 40-60%, (5) 60-80%, and (6) 80-100%." However, it is not clear how they weighted the different land uses considered, whose ecological impacts and thus their "intensity" clearly differ. For instance, we cannot assume that a rangeland area, which can be managed in a sustainable way in drylands and has impacts that largely depend on grazing intensity and other biotic and abiotic factors (see for instance <https://www.science.org/doi/10.1126/science.abq4062>, <https://www.sciencedirect.com/science/article/abs/pii/S1877343520300798>), has the same ecological

impact or intensity than an urban area. In the same way, the impacts of agriculture largely depend on whether it is rainfed or irrigated, if it is conventional, regenerative, or organic and so on. Any attempt to measure land use intensity without accounting for the differential impacts of different land uses will lead to misleading results. As an example, and using the baseline (2015) scenario, the authors report that large parts of Saudi Arabia have a high land use intensity and thus are largely impacted by human activities, something that it is difficult to believe. The same can be said to other dryland regions. Given the high land use for many drylands in the baseline map used, it is not surprising to find that it will worsen in the future regardless of the scenarios considered.

Because the way the authors defined and quantified land use intensity is critical for trusting the results and bold take-home messages provided in this manuscript, the authors should fully justify their approach to convince critical readers (like me!) about the suitability of the approach they followed, and thus about the validity of the interpretation the authors are making from their results. Also, and given the important implications of the results presented, I feel the authors should also validate their land use intensity classification using independent data. This would not only substantially strength the manuscript but if this is not done then I am afraid that the results presented cannot be trusted.

A couple of minor comments are below:

- 1) While the figure is simple, I had problems to fully understand Fig. 2 based on the information provided in the caption. Please rewrite it to allow readers to fully understand what this figure shows.
- 2) L411-412. The mention to "Presumed Drylands" is confusing, either rewrite to clarify or delete it

In summary, my impression is that, albeit this study deals with an interesting and very important topic, and provides important and far-reaching results, some of them, and particularly those from the section should be viewed with some caution given the potential limitations of the methodological approach followed by the authors. To further improve this study I would advise the authors to: i) remove -or better justify and discuss- the first part and, most importantly, ii) to justify and fully validate using independent data the methodology used to quantify land use intensity, and iii) place and interpret their results given the limitations of the methodological approach used by the authors.

I hope that these criticisms, which I have made with the idea to be constructive (please let the editor know if you think they are harsh or inadequate), will be useful to further revise and improve this work.

Reviewer #2 (Remarks to the Author):

The central thesis of this manuscript is threats to global drylands. To articulate this, the authors bring together three, in my view, poorly connected concepts; the lack of dryland definitions, the coverage of protected areas, and an assessment of threats status of four vertebrate groups. I have no doubt that there are potential problems with different methods of classifying drylands, but in the context of threats, it is unclear how this particular analysis is relevant to a discourse on threats. When one think about threats, one generally thinks about phenomena such as climate change, land-use conversion, deforestation, desertification, population change, woody plant invasion, weed invasions etc as well as the many social and political issues that threatens the viability, stability and function of drylands. This manuscript tries to be about threats, but there is little discussion of threats, nor is it clear how the examination of different mapping systems addresses the concept of threats. Rather than just a mapping exercise, the reader will want to know more about the threats, ecological and otherwise.

In their opening paragraph, the authors suggest that the lack of a consistent definition is a major obstacle to conservation planning. I'm not sure how they justify this. The literature is replete with examples of threats to drylands. The lack of consistent mapping does not appear as a major threat to conservation planning in drylands. Even if this were true and that the definitions and spatial extent of

different classification systems were an obstacle, there is still relatively strong concordance between the different systems, so it's hard to see how this is an obstacle.

The section on protected areas and biodiversity threats across drylands is really interesting, but being largely descriptive, misses some depth. One is left wondering about the links between percent protected areas and conservation. This could be made more explicit in the manuscript with more linking to ecology and ecological processes. Similarly, statements about protected areas permitting multiple land uses and resource extraction are not particularly novel. More information could be provided here to guide the reader to exactly what these authors were referring to.

The material on pages 11 and 12 is extremely valuable and this is where the strength of the manuscript lies. This section aims to explore the threats to biodiversity by looking at four different faunal groups. It is not clear how the authors chose 50% cut off. This differs from their 99% cut off for endemic species. 50% seems very broad to me and I wonder how values would be different if say they chose 90%, which would seem more realistic in terms of conservation. It would have been really good to have had a discussion of some of the broad ecological mechanisms underpinning these threats rather than just a delineation of the spatial and aerial aspects.

There is a lot of potential for looking at biodiversity threats in drylands, and it's good to see that the authors have uncovered major differences in our definitions of drylands.

Overall, I had the impression that the authors were attempting to force fit three different work packages into a coherent framework under a discussion of threats, but unfortunately it is not particularly convincing.

Reviewer #3 (Remarks to the Author):

10 October. First, deepest sympathies with what your country is currently experiencing!

I have reviewed Lewin et al.'s manuscript titled "Current and future threats to global drylands". In this manuscript, the authors first show overlaps and differences between four different global maps of dryland. They then proceed to examine how one of these layers (the results from the other three are presented in the supplementary material): 1) overlap with protected areas, 2) represent biodiversity (using amphibians, birds, mammals, and reptiles), and 3) look at how dryland overlaps with projected land cover changes in 2100. As the authors also point out, drylands are often overlooked in conservation and, thus, this paper provides an important (mostly descriptive) contribution to the conservation literature that I believe will be relevant for a broad audience, and which I believe falls very well within the scope of the journal. The manuscript is well written in a clear English and I overall only

have a few comments:

I think the authors are missing an opportunity to do more on their analysis comparing the four layers. A more in-depth compare-and-contrast analysis that looks into what environmental and human factors describe the differences would be very interesting. I realize this could likely be its own paper and might be too much for this manuscript (that is likely for the editor and authors to decide), but I did feel that the first part of the results; simply looking at where the four measures overlap or not, left me with more questions than answers – and questions I believe would not be too hard to answer using additional data on climate, human pressure, land use, habitat classifications etc.

I think the text on the four different layers (ca. lines 74-89) would be much better represented by a table that outlines the main aspects of the four layers. I would also make it much easier to highlight what are the important differences in columns (i.e. source data, resolution, types drylands). I think this would aid the reader and allow for the authors to structure their main points about the data more

transparently

Presenting results in a journal that has (for reasons that are beyond my comprehension) chosen to put the methods after the results, requires I think, a bit more indirect introduction of the methods in the results. For example in the sentence starting on line 126, I would suggest the addition of the data (i.e. the World Database on Protected Areas) be included. This could be like: "Using the World Database on Protected Areas [REF] we show that drylands are considerably less covered by the global network of protected areas compared to 127 non-drylands (Fig. 2a)." this is just a suggestion, but I think the results section, more generally, would benefit from these types of small additions that makes it a little less necessary to flip between the methods and results when reading.

Figure 2: the description of the white portion of the graph seems misleading in the figure legend. It currently reads "white = all IUCN categories I-VI.", but isn't it the sum of the color and the white that is all IUCN categories – the white part alone is only categories V and VI?

The use of thresholds of protection (i.e. 10%) is a bit crude and doesn't include long established approaches to integrate the total range-size of a species when assessing adequacy of protection. There is nothing wrong in what the authors do, but it would be more informative and align better with conventions of how this is reported, if the authors used a measure of adequacy rather than an arbitrary 10% threshold. I would suggest the authors consult Rodrigues, et al. 1, to see the methods used there, which would allow the authors to use more meaningful measures of 1) the proportion of species with no protection, 2) the proportion with adequate protection, and 3) the proportion with some but not adequate protection. It might also be worth looking at Butchart, et al. 2 for a more recent version.

In the use of the WDPA, its important to realize that the database has some major gaps. Some countries don't allow for their data to be displayed and downloaded. This can change with the specific version, but from 2017/18 this has been China and more recently (but probably before 2022) India and a few other countries. If all that has been done in terms of the protected area results is some filtering based on IUCN management categories the results might be misleading for reasons that simply has to do with what data has been used. The authors will need to check for any systematic gaps in the data and incorporate this in the results. This should be possible to see by simply visually inspecting the data on a map. Alternatively, this should be reported in the notes to the specific version downloaded. This will likely means re-running all protected area analysis.

Minor comments

Lines: 71-72: Not sure what this sentence means

Line 72: should "compared" be present tense (i.e. "compare")

Line 137: strictness (i.e. strictly for biodiversity) in relation to the IUCN categories is a complicated discussion. For example, category VI have in many cases been shown to outperform other "lower" level/numbered categories. And indeed, in particular category IV, but also III, and II allows for other considerations than biodiversity. Though I would personally never divide the IUCN categories into I-IV and V-VI, I think the division can be accepted, but the language around it needs to be refined.

Lines 213-214 and Lines 224-225: in the first case the authors write: "... (SSP2-4.5), 100% of drylands are projected to be converted to human land uses" and in the second "... (SSP5-8.5), most natural drylands are projected to undergo some degree of conversion to human land-uses". I am a little confused as to how 100% was converted under the less depressing scenario and "only" most was projected to "undergo some degree of conversion". It might just be the authors trying to not make the text too "dry"/repetitive, but it actually opens up room for some misunderstandings.

Line 422: I just need to be sure, you were working at a 0.002 km² resolution in a global analysis? Or what is this?

Lines 436-442: its not entirely clear to me from the description here how protected areas that was not categorized as one of the six/seven IUCN categories were treated. I don't have strong opinions about how they should be used (or not used) but clarity about what was done is important.

1 Rodrigues, A. S. L. et al. Global gap analysis: Priority regions for expanding the global protected-area network. *BioScience* 54, 1092-1100 (2004).

2 Butchart, S. H. M. et al. Shortfalls and solutions for meeting national and global protected area targets. *Conservation Letters* 8, 329-337 (2015).

*****END*****

Author Rebuttal to Initial comments

RESPONSE TO REVIEWER #1

COMMENT

In this manuscript, the authors “evaluated four commonly used global delineations of drylands to assess their overlap, the formal degree of protection of global drylands, their land-vertebrate biodiversity, and projected human-induced threats to drylands under different future climate change and socioeconomic scenarios”. This work deals with a very interesting topic, which is undoubtedly of interest to Nature Ecology and Evolution readers. The paper was also very well written and was an enjoyable read.

Response:

Thank you!

I value the attempt made by the authors and I found the results potentially far-reaching. However, at the same time I am skeptical with some of the take-home messages given by the authors because of the limitations of the methodological approach they used (see specific comments below). I will comment below on the three main sections of the paper.

Regarding the Dryland delineations section, my feeling is that this section is somewhat disconnected from the rest and it is not fully clear the value it adds to the manuscript. On the one hand I have the feeling that the authors are somehow artificially trying to make the case for disagreements and debates that I feel that are not dominant when they include sentences such as

“A major obstacle for conservation planning in drylands is the lack of a consistent definition of what constitutes drylands”, “There is much overlap between these delineations, without a widely accepted definition to categorize drylands. This lack of clarity can greatly hamper broad-scale conservation efforts in drylands” or “Such incongruencies among dryland delineations may have important implications for guiding scientific discourse, land management, and identifying broad-scale dryland conservation priorities.” To me the UNEP definition is widely used and accepted, both within the scientific and policy communities, and thus is not as debated as suggested. Indeed, it is not surprising to find that is the definition used by the authors in the analyses they present in the text. But most importantly, my feeling is that in this exercise, while interesting, the authors are trying to compare classifications that are not fully comparable. Of course, one can expect to find large differences if we compare a classification based purely on climatic variables (such as the UNEP one) with other classification based on biomes or agricultural potential, such as the WWF or FAO ones. Different classifications have different objectives and I was not convinced that the analyses presented in the first part of the paper, while undoubtedly interesting, add anything relevant to the paper. Furthermore, if the incongruencies between the classifications mentioned in sentences such as those highlighted above are really problematic for delineating conservation/management actions then this should be reflected in the results of the analyses done by the authors, right? But this does not seem to be the case as reported by the authors (“Parallel analyses for the other delineations are provided in Extended Data and Supplementary Table S1, revealing similar trends as described below.”). Thus, if the authors want to retain this part of the manuscript, I think they should provide additional justification of the value, soundness and need of the comparison among classifications they did.

Response:

We appreciate this comment, and as the remaining reviewers made similar comments regarding the relevance of this section comparing dryland delineations, we have decided to remove it. In our revised manuscript we focus only on the UNEP-WCMC drylands dataset as the reviewer suggests. Please see the section titled “Global drylands” where we describe the dataset briefly, and Fig. 1.

Lines 66-82: *“We evaluated drylands as classified by the UNEP-WCMC (United Nations Environment World Conservation Monitoring Centre)¹. Drylands are categorized using an aridity index (the ratio of annual precipitation to potential evapotranspiration), with values below 0.65. Drylands are further divided into hyperarid, arid, semiarid, and dry subhumid subtypes (Fig. 1). This drylands designation dataset is commonly referred to by the Convention of Biological Diversity (CBD)³⁴ and United Nations Convention to Combat Desertification (UNCCD)³⁵ for policy and management goals, and often cited in the scientific literature (e.g., ref.⁸). Drylands cover about 42% of earth’s terrestrial surface and encompass diverse and unique regions globally (Fig. 1). Most dryland regions are found in Africa and Asia, which combined encompass 64% of global drylands and also comprise the largest areas of hyperarid and arid subtypes (Extended Data Table 1). Drylands are mostly inclusive of the Deserts & Xeric Shrublands Biome^{10,11}, and also comprise large proportions of other biomes and diverse*

habitats³ across continents. These range from subhumid zones (e.g., Tropical & Subtropical Grasslands, Savannas & Shrublands) especially in regions of Africa and Australia (Fig. 1 and Extended Data Fig. 1), to Mediterranean regions of Europe, temperate zones (e.g., Temperate Grasslands, Savannas & Shrublands) especially in Asia and North America, and small portions of sub-polar regions (e.g., Boreal Forests/Taiga and Temperate Conifer Forests)."

COMMENT

I really liked the second part, Protected areas and biodiversity threats across drylands, which presents very interesting and relevant findings in a very clear manner. Nothing to object to this part.

Response:

Thank you!

COMMENT

I am, however, particularly concerned about the analyses and messages given in the third part of the manuscript, Current and projected anthropogenic threats to drylands. First, I have to say that I am not sure to have fully understood the analyses they did because the methodological part of the manuscript is very brief and do not provide all the explanations/details needed to allow readers to fully understand what they did.

Response:

We have attempted to clarify and describe these changes below.

Regardless of this important point, I am particularly worried with the land use intensity classification made by the authors, which is not properly explained/justified and indeed deliver results that are difficult to trust. If I properly understood what the authors did, they quantified the cover of the following land uses: 'managed pasture', 'rangeland', 'urban land', 'C3 annual crops', 'C3 perennial crops', 'C4 annual crops', 'C4 perennial crops' and 'C3 nitrogen-fixing crops' and then "overlaid all land-cover layers with drylands and identified the proportional cover of land cover types per pixel, which we then classified to the following land-use intensity categories to help with interpretation: (1) 0%, (2) 0-20%, (3) 20-40%, (4) 40-60%, (5) 60-80%, and (6) 80-100%."

Response:

This is correct. The Land Use Harmonization dataset version 2 (LUH2) describes the proportional land use of 0.25° grid cells. We focus on four primary land use categories (cropland,

pasture, rangeland, and urban land) – representing all non-natural land uses excluding natural vegetation (i.e., primary or secondary forest or non-forest). The map in Fig. 6 (provided below) shows the fraction of each grid cell occupied by these land types combined. **The circles show the total land cover area of grids occupied by some fraction of the above land types, and additional fractional land areas occupied under future projected scenarios.**

However, it is not clear how they weighted the different land uses considered, whose ecological impacts and thus their “intensity” clearly differ. For instance, we cannot assume that a rangeland area, which can be managed in a sustainable way in drylands and has impacts that largely depend on grazing intensity and other biotic and abiotic factors (see for instance <https://www.science.org/doi/10.1126/science.abq4062>, <https://www.sciencedirect.com/science/article/abs/pii/S1877343520300798>), has the same ecological impact or intensity than an urban area. In the same way, the impacts of agriculture largely depend on whether it is rainfed or irrigated, if it is conventional, regenerative, or organic and so on. Any attempt to measure land use intensity without accounting for the differential impacts of different land uses will lead to misleading results. As an example, and using the baseline (2015) scenario, the authors report that large parts of Saudi Arabia have a high land use intensity and thus are largely impacted by human activities, something that it is difficult to believe. The same can be said to other dryland regions. Given the high land use for many drylands in the baseline map used, it is not surprising to find that it will worsen in the future regardless of the scenarios considered.

Because the way the authors defined and quantified land use intensity is critical for trusting the results and bold take-home messages provided in this manuscript, the authors should fully justify their approach to convince critical readers (like me!) about the suitability of the approach they followed, and thus about the validity of the interpretation the authors are making from their results.”

Response:

We do not weigh the different potential impacts of different land classes and their land uses (but we discuss some of these now in the main text), instead we assign an equal value to each land class. Such impacts are challenging to weigh globally due to regional variations and sensitivity of different animals and plants to different pressures, moreover other studies that evaluated land use impacts did not apply such weighting for similar reasons (e.g., Powers et al., 2019 - <https://doi.org/10.1038/s41558-019-0406-z?>).

We attempt to clarify this better now in the Methods:

Line 575-585: “We overlaid land cover layers with drylands and identified the land-use fractions of grid cells of exactly 0.22180° resolution (projected using an equal-area Behrmann projection). Grid cells were determined as dryland if grid cell centers fall within the dryland polygon layer. We classified grids according to the following intervals to help with

interpretation: (1) 0%, (2) 0-20%, (3) 20-40%, (4) 40-60%, (5) 60-80%, and (6) 80-100%. We treated each land class equally without applying different weights⁵⁶ because impacts are challenging to weigh globally due to non-stationarity and regional variations and sensitivities of different animals and plants to different pressures. To calculate projected future land area of natural drylands converted, we summed the total land cover area of grids occupied by some fraction of the above land classes (i.e., all grids above 0% land-use fraction) – herein ‘total fractional land areas’, plus the additional total fractional land areas occupied under future projected SSPs (i.e., circles in Fig. 6).”

It is important to clarify the difference between fractional land cover (i.e., proportion of a grid cell occupied by a land use type) as we refer to, and land-use intensity in terms of the intensity of inputs and productivity of managed lands. The reviewer is correct in stating that rangeland might not be as ecologically intensive as other forms of land use (Goldewijk et al., 2017; <https://essd.copernicus.org/articles/9/927/2017/>), for example irrigated cropland. Indeed, “rangelands” as defined here, comprise a variety of ecosystems – natural and unmanaged habitats including grasslands, shrublands, woodlands and deserts containing native vegetation, and typically have low livestocking densities. **Therefore, the high fraction of land use occupied in Saudi Arabia for example, as the reviewer indicates (in Fig. 6 below), is mostly due to expansive areas of rangelands occupying large fractions of natural dryland.** We do not imply that these lands are managed more intensively than other land types (e.g., cropland or built-up areas).

Following the reviewer’s comment, we investigated further the contribution of rangelands to overall land cover. We repeated our analysis with croplands, pasture, and urban lands – but excluded rangelands as a converted land-cover type (Extended Data Fig. 7 shown below). The baseline map from 2015 (Extended Data Fig. 7a) shows fewer reduced areas of fractional land cover (as the reviewer suspects) compared to Fig. 6a. **Nevertheless, the large increases in fractional land cover under projected scenarios persist, mostly due to the considerable amounts of drylands converted to croplands and urban areas.** For example, in the SSP1-2.6 scenario – **95% of drylands** are projected to be converted to some degree including all land classes (Fig. 6b), compared to **92% of drylands** converted to some degree when excluding rangeland (Extended Data Fig. 7b) – a negligible difference.

Fig. 6. Current land-use patterns in drylands, LUH2 (2015) (a), and under shared socioeconomic pathways (SSPs), SSP1-2.6 (2100) (b), SSP2-4.5 (2100) (c), and SSP5-8.5 (2100) (d). Colour codes correspond to fraction of

nature portfolio

grid cell occupied by cropland, pasture, rangeland, and urban land classes from low (yellow) to high (blue). Circles show total land cover area of 25 km² grids occupied by some fraction of land classes above 0% (i.e., total fractional land areas). Pink = total fractional land areas in 2015 (a), purple = projected additional total fractional land areas (b-d).

Extended Data Fig. 7. Current land-use patterns in drylands LUH2 (2015) (a), and under shared socioeconomic pathways (SSPs), SSP1-2.6 (2100) (b), SSP2-4.5 (2100) (c), and SSP5-8.5 (2100) (d). Colour codes correspond to fraction of grid cell occupied by cropland, pasture, and urban land classes excluding rangeland

from low (yellow) to high (blue). Circles show total land cover area of 25 km² grids occupied by some fraction of land classes above 0% (i.e., total fractional land areas). Pink = total fractional land areas in 2015 (a), purple = projected additional total fractional land areas (b-d).

Additionally, in Extended Data Figs. 8-10 (shown below) we show trends in the gain or loss of land cover fractions per grid for each land type relative to the baseline (LUH2). Croplands and urban lands show considerable increases in the number of grids with increasing land cover fractions (i.e., “gains”) compared to rangelands and pasture (i.e., intensively managed grazing lands) under any scenario. This further indicates that large areas of natural drylands are projected to be converted, especially as a result of increasing cropland and urban areas. Additionally, we show that large areas of rangelands (including native shrublands, deserts, etc.) are predicted to be converted as well. **Consequently, we think that these combined results (presented in Extended Data Figs. 7-10) allow us to reliably interpret the projected large gains of converted drylands in Fig. 6.** Following this important comment by the reviewer, we have now added to our manuscript these new analyses and refer to it in the Main Text, for example:

Lines 218-231: *“Again, most natural drylands are projected to be converted by some fraction under such a scenario (9.6 M km² additional natural drylands will be converted; Fig. 6b), especially as a result of increases in cropland and urban areas with concomitant reductions in pasture and rangeland (including unmanaged grasslands and shrublands with native vegetation)⁶⁰ – an apparent trend across different socioeconomic pathways (Extended Data Figs. 7-10). Therefore, large areas of natural drylands are projected to be converted as a result of increasing cropland and urban areas, while large areas of rangelands (including native shrublands, deserts, etc.) are predicted to be converted as well. This is mostly a consequence of the vast land resources needed for agriculture⁸ and alternative energy production (i.e., solar panels and bioenergy)⁶¹. Moreover, the development of solar energy resources has wide-ranging impacts on dryland ecosystems and biodiversity through habitat fragmentation and loss⁶². Also, the significant expansion of urban land types and infrastructure in drylands including artificial structures and surfaces, roads and mining/extraction sites^{63,64} are expected to have extensive impacts on dryland species⁶⁵⁻⁶⁷ (Fig. 5).”*

We further added details on this new exploration in the Methods:

Lines 586-591: *“Rangelands comprise vast areas of natural and unmanaged habitats and ecosystems including grasslands, shrublands, woodlands and deserts containing native vegetation, and typically have low livestock densities^{40,60} – possibly exerting a disproportionate influence on the land-use patterns reported. Therefore, to quantify the relative contribution of rangeland to total fractional land areas, we repeated the above analysis for cropland, pasture, and urban land classes combined excluding rangelands (Extended Data Fig. 7).”*

Extended Data Fig. 8. Change of land cover fraction of grids occupied by land types: cropland (a), pasture (b), rangeland (c), and urban land (d) relative to current values LUH2 (2015) under shared socioeconomic pathway (SSP), SSP1-2.6 (2100). Blue = gain in land cover fraction, tan = loss, light green = no change.

nature portfolio

Extended Data Fig. 9. Change of land cover fraction of grids occupied by land types: cropland (a), pasture (b), rangeland (c), and urban land (d) relative to current values LUH2 (2015) under shared socioeconomic pathway, SSP2-4.5 (2100). Blue = gain in land cover fraction, tan = loss, light green = no change.

Extended Data Fig. 10. Change of land cover fraction of grids occupied by land types: cropland (a), pasture (b), rangeland (c), and urban land (d) relative to current values LUH2 (2015) under shared socioeconomic pathway, SSP5-8.5 (2100). Blue = gain in land cover fraction, tan = loss, light green = no change.

COMMENT

Also, and given the important implications of the results presented, I feel the authors should also validate their land use intensity classification using independent data. This would not only substantially strength the manuscript but if this is not done then I am afraid that the results presented cannot be trusted.

Response:

We now compare the LUH2 dataset applied here to drylands with a recently published dataset – the Future Land Use Simulation model (FLUS) by Chen et al (2022). This dataset is downscaled to a finer resolution, therefore we cannot compare the fractional land covers directly, however when comparing global values of total land areas (10^6 km) of comparable land types in drylands under projected future scenarios, **we find that the general trajectories are similar to LUH2 (please see Response Fig. 1 below) in drylands.** See also Zhang et al. (2023) and the Global LULC projection dataset, showing a high degree of spatial consistency and correlations among datasets. **These datasets validate the LUH2 dataset we use (please see Response Figs. 2 and 3 below).**

Chen, G., Li, X. & Liu, X. Global land projection based on plant functional types with a 1-km resolution under socio-climatic scenarios. *Sci Data* 9, 125 (2022) <https://doi.org/10.1038/s41597-022-01208-6>

Zhang, Tianyuan; Cheng, Changxiu; Wu, Xudong (2023). Global LULC projection dataset from 2020 to 2100 at a 1km resolution. figshare. Dataset. <https://doi.org/10.6084/m9.figshare.23542860.v1>

Response Fig. 1. We compared the demands for major land types in drylands between datasets LUH2 and FLUS (from Chen et al., 2022) under SSP scenarios. These show comparable trends. Lines show differences in land area (10⁶ km) between years 2015 and 2100 for scenario SSP1-2.6 (2100) in blue, SSP2-4.5 (2100) in green, and SSP5-8.5 (2100) in red.

nature portfolio

Response Fig. 2 (from Chen et al., 2022; Fig. 4). “Comparison of demands for major land types between ours and LUH2 under the SSP-RCP scenarios on a global scale (2015–2100).” <https://doi.org/10.1038/s41597-022-01208-6>.

Response Fig. 3 (from Zhang et al., 2023; Fig. 10). “Comparison of the simulated LULC result of a typical case region (within North America) in 2100 under the SSP4-3.4 scenario between the dataset produced in this study and LUH2.” <https://doi.org/10.6084/m9.figshare.23542860.v1>

COMMENT

A couple of minor comments are below:

1) While the figure is simple, I had problems to fully understand Fig. 2 based on the information provided in the caption. Please rewrite it to allow readers to fully understand what this figure shows.

Response:

Reviewer #3 made a similar comment, therefore we simplify Fig. 2a by including non-drylands directly, instead of showing dashed lines, and clarify the caption. Please see Fig. 2 and the figure caption (shown below).

COMMENT

2) L411-412. The mention to “Presumed Drylands” is confusing, either rewrite to clarify or delete it.

Response:

We rewrite this here, hopefully this is clearer now.

Lines 465-473: *“We evaluated the geographic extent of drylands as classified by the United Nations Environment World Conservation Monitoring Centre (UNEP-WCMC)¹ using the ‘Drylands Dataset (2007)’. Drylands are categorized based on aridity values defined as the ratio of precipitation to potential evapotranspiration (P/PET), with values ranging below 0.65. Drylands are further distinguished between subtypes using aridity intervals – hyperarid (<0.06 P/PET), arid (0.06-0.2 P/PET), semiarid (0.21-0.5 P/PET), and dry subhumid (0.51-0.65 P/PET). We excluded ‘Presumed Drylands’ – additional areas that do not fall within the above intervals (i.e., P/PET ≥0.65) but may contain “dryland features” – data downloaded from [12](https://datadownload.unep-wcmc.org/datasets.”). The LUH2 dataset integrates historical human land use data, management activities, and maps with projected models simulating future land-use changes under different scenarios. These models comprise a number of sub-models describing agricultural, demographic, socioeconomic, vegetation and climate systems operating at several different spatial resolutions (e.g., local, regional, national)⁶¹. Therefore, the harmonization strategy estimates fractional land-use patterns at a globally consistent scale of 0.25° grids integrating historical reconstructions with future projections of land use incorporating multiple sub-datasets. Recently released datasets at higher resolutions – see ref. ⁷⁷ (e.g., Fig. 10), ref. ⁷⁸ (e.g., Fig. 4), ref. ⁷⁹) – correlate strongly with LUH2 spatially for comparable land classifications, however these datasets may have several limitations based on data availability yielding potential errors in future land projections. Also, these datasets do not distinguish between grazing areas (i.e., rangeland, grassland and pasture regions), which have particular relevance and distinctions in drylands. Therefore, we use LUH2 – widely applied in spatially explicit global land-use analyses (e.g., refs. ^{80–82}). However, the data used contain several limitations including:*

the coarse scale, which may cause some deviations in land cover patterns and prevent local analyses, regional differences in estimating future land-use changes based on historical data availability, and additional uncertainties on biodiversity impacts regarding species-specific responses and sensitivities to different land types based on behavioural and physiological traits^{31,83}.”

COMMENT

I hope that these criticisms, which I have made with the idea to be constructive (please let the editor know if you think they are harsh or inadequate), will be useful to further revise and improve this work.

Response:

Very constructive indeed. Thank you again!

RESPONSE TO REVIEWER #2

COMMENT

The central thesis of this manuscript is threats to global drylands. To articulate this, the authors bring together three, in my view, poorly connected concepts; the lack of dryland definitions, the coverage of protected areas, and an assessment of threats status of four vertebrate groups. I have no doubt that there are potential problems with different methods of classifying drylands, but in the context of threats, it is unclear how this particular analysis is relevant to a discourse on threats. When one thinks about threats, one generally thinks about phenomena such as climate change, land-use conversion, deforestation, desertification, population change, woody plant invasion, weed invasions etc as well as the many social and political issues that threaten the viability, stability and function of drylands. This manuscript tries to be about threats, but there is little discussion of threats, nor is it clear how the examination of different mapping systems addresses the concept of threats. Rather than just a mapping exercise, the reader will want to know more about the threats, ecological and otherwise.

In their opening paragraph, the authors suggest that the lack of a consistent definition is a major obstacle to conservation planning. I'm not sure how they justify this. The literature is replete with examples of threats to drylands. The lack of consistent mapping does not appear as a major threat to conservation planning in drylands. Even if this were true and that the definitions and spatial extent of different classification systems were an obstacle, there is still relatively strong concordance between the different systems, so it's hard to see how this is an obstacle.

Response:

Thank you for these comments. The remaining reviewers have made similar comments regarding the relevance of this section comparing dryland delineations, therefore we have decided to remove it, instead focusing only on the UNEP-WCMC drylands dataset as Reviewer #1 suggests. Please see the section titled "Global drylands" where we describe the dataset briefly, and Fig. 1.

Lines 66-82: *"We evaluated drylands as classified by the UNEP-WCMC (United Nations Environment World Conservation Monitoring Centre)¹. Drylands are categorized using an aridity index (the ratio of annual precipitation to potential evapotranspiration), with values below 0.65. Drylands are further divided into hyperarid, arid, semiarid, and dry subhumid subtypes (Fig. 1). This drylands designation dataset is commonly referred to by the Convention of Biological Diversity (CBD)³⁴ and United Nations Convention to Combat Desertification (UNCCD)³⁵ for policy and management goals, and often cited in the scientific literature (e.g., ref.⁸). Drylands cover about 42% of earth's terrestrial surface and encompass diverse and unique regions globally (Fig. 1). Most dryland regions are found in Africa and Asia, which combined encompass 64% of global drylands and also comprise the largest areas of hyperarid and arid subtypes (Extended Data Table 1). Drylands are mostly inclusive of the Deserts & Xeric Shrublands Biome^{10,11}, and also comprise large proportions of other biomes and diverse*

habitats³ across continents. These range from subhumid zones (e.g., Tropical & Subtropical Grasslands, Savannas & Shrublands) especially in regions of Africa and Australia (Fig. 1 and Extended Data Fig. 1), to Mediterranean regions of Europe, temperate zones (e.g., Temperate Grasslands, Savannas & Shrublands) especially in Asia and North America, and small portions of sub-polar regions (e.g., Boreal Forests/Taiga and Temperate Conifer Forests)."

We agree with the reviewer's comment regarding threats such as climate change, land-use conversion, deforestation, desertification, etc. and other ecological processes. **We have now added a section under "Threats to biodiversity in drylands and conservation gaps", in which we assess the specific types of threats (using IUCN threats classification data) impacting dryland land vertebrate species, and we add another figure.** We find that agriculture and urban infrastructure are particularly significant threats (among other threats), which we also show are projected to expand considerably under future land-use scenarios. Please see Fig. 5 (provided below), and:

Lines 165-178: *"We further explored threat types currently affecting dryland species as designated by the IUCN threat-classification scheme per species (v.3.3; Fig. 5). The largest and most significant anthropogenic threat to vertebrate groups in drylands is agriculture, as reported in other regions⁴⁸⁻⁵⁰ (Extended Data Fig. 6). Other prevalent threats in drylands include: timber and plant harvesting (e.g., harvesting plants and trees for commercial, recreational, and subsistence uses), threats from invasive species and disease, and infrastructure development. However, different threats emerge as important across different groups in drylands. For example, amphibians are the most impacted by water management in drylands (16% of species assessed) compared to the other groups (Fig. 5a), climate change is a larger threat to birds (which are especially vulnerable to heatwaves)³² in drylands (29% of species assessed; Fig. 5b), over-exploitation (i.e., direct hunting and harvesting of animals) is a greater threat to mammals (42% of species assessed; Fig. 5c), and infrastructure development (i.e., residential/commercial development and transportation/service corridors) and mining/energy production threaten reptiles more than the other groups (37% and 21% of reptile species assessed, respectively; Fig. 5d)."*

We elaborate on the methods used under the Methods section:

Lines 529-538: *"We further assessed the specific types of threats affecting dryland species in IUCN threatened categories (VU, EN, CR; Fig. 5) using the IUCN threat-classification scheme v.3.3 – providing comprehensive data on known threats per species (data obtained from www.iucnredlist.org). We grouped similar threats to allow for simpler comparison according to ref. ⁴⁸, except for the grouping of 'Invasive species', which we we combined with diseases under the grouping 'Invasion and disease' as according to ref. ⁴⁹. When relevant, multiple threats were coded per species (e.g., 'Agriculture' and 'Over-exploitation'). Following ref. ⁴⁹, we considered*

only future and ongoing threats, we omitted threats affecting only a minority of the global population (i.e., <50% of the population) per species, and we removed 'negligible' threats and those causing 'no declines'.

[Partially Redacted]

Fig. 5. Types of threats affecting amphibian (a), bird (b), mammal (c), and reptile (d) threatened species (in IUCN categories CR, EN, VU) in drylands ($\geq 50\%$ global range in drylands). Percentages show the proportion of species assessed affected by threats per taxon. Most species are subjected to multiple threats.

COMMENT

The section on protected areas and biodiversity threats across drylands is really interesting, but being largely descriptive, misses some depth. One is left wondering about the links between percent protected areas and conservation. This could be made more explicit in the manuscript with more linking to ecology and ecological processes. Similarly, statements about protected areas permitting multiple land uses and resource extraction are not particularly novel. More information could be provided here to guide the reader to exactly what these authors were referring to.

Response:

We aimed to refer to various specific threats that may affect the different groups of land vertebrates (as referred to above). As we are dealing with thousands of species globally these are broad generalizations, but we do provide stronger links to the land types examined, for example:

Lines 227-233: *“Moreover, the development of solar energy resources has wide-ranging impacts on dryland ecosystems and biodiversity through habitat fragmentation and loss⁶². Also, the significant expansion of urban land types and infrastructure in drylands including artificial structures and surfaces, roads and mining/extraction sites^{63,64} are expected to have extensive impacts on dryland species⁶⁵⁻⁶⁷ (Fig. 5). Such threats are likely to have multiple and interactive effects on dryland species further exacerbated by climate change and habitat fragmentation⁶⁸.”*

And in the Abstract:

Lines 23-33: *“Consequently, most dryland vertebrates including many endemic and narrow-ranging species are inadequately protected (0-2% range coverage). Dryland vertebrates are threatened by varied anthropogenic factors – including agricultural and infrastructure development (i.e., artificial structures, surfaces, roads and mining/extraction sites). Alarming, by 2100 drylands are projected to experience some degree of land conversion in 95-100% of current natural habitat due to urban, agricultural, and alternative energy expansion. This loss of undisturbed dryland regions is expected across different socioeconomic pathways, even under optimistic scenarios characterized by progressive climate policies and moderate socioeconomic trends. Ultimately, current and future threats to global drylands demand urgent conservation policies and actions to safeguard these unique and invaluable habitats.”*

COMMENT

The material on pages 11 and 12 is extremely valuable and this is where the strength of the manuscript lies. This section aims to explore the threats to biodiversity by looking at four different faunal groups. It is not clear how the authors chose 50% cut off. This differs from their 99% cut off for endemic species. 50% seems very broad to me and I wonder how values would

be different if say they chose 90%, which would seem more realistic in terms of conservation. It would have been really good to have had a discussion of some of the broad ecological mechanisms underpinning these threats rather than just a delineation of the spatial and aerial aspects.

Response:

Thank you, and please see our response above where we elaborate on threats and linkages to anthropogenic factors. We now repeat the analyses in this section also for species with $\geq 90\%$ of their global ranges in drylands, and we find similar trends as reported in the Main Text for species with $\geq 50\%$ of their distributions occurring in drylands (please see Supplementary Figs. S2-S4 below). These more dryland specialist species (with $\geq 90\%$ of their global ranges in drylands) have larger proportions with either 0% protection or 10-100% protection. This is to be expected as these species have narrower ranges than our original subset and thus either fall completely outside or predominantly within PAs. Moreover, as this new subset has narrower ranges they are also more threatened (according to the IUCN Red List) than our original subset of species with at least 50% of their range in drylands.

Lines 519-521: *“We repeated the above analyses for species with $\geq 90\%$ of their global ranges in drylands, and find similar trends as reported above for species with $\geq 50\%$ of their distributions occurring in drylands (Supplementary Figs. S2-S4).”*

[Partially Redacted]

Supplementary Fig. S2. Proportion range size protected and IUCN threat status of amphibian (a), bird (b), mammal (c), and reptile (d) species in drylands ($\geq 90\%$ global range in drylands). Inner circles show proportion of species with range size protected (IUCN categories I-IV): orange = 0% range protected, blue = 0-10% range protected, green = 10-100% range protected. Outer circles show proportion of IUCN threatened species: dark orange = threatened (IUCN categories CR, EN, VU), dark green = non-threatened (IUCN categories NT, LC), grey = data deficient (IUCN category DD). n = number of species, parentheses = number of endemic species ($>99\%$ range in drylands).

[Partially Redacted]

Supplementary Fig. S3. Proportion range size protected and IUCN threat status of amphibian (a), bird (b), mammal (c), and reptile (d) species by continent in drylands ($\geq 90\%$ global range in drylands). Inner circles show proportion of species with range size protected (IUCN categories I-IV): orange = 0% range protected, blue = 0-10% range protected, green = 10-100% range protected. Outer circles show proportion of IUCN threatened

species: dark orange = threatened (IUCN categories CR, EN, VU), dark green = non-threatened (IUCN categories NT, LC), grey = data deficient (IUCN category DD). n = number of species, parentheses = number of endemic species (>99% range in drylands).

[Partially Redacted]

Supplementary Fig. S4. Types of threats affecting amphibian (a), bird (b), mammal (c), and reptile (d) threatened species (in IUCN categories CR, EN, VU) in drylands ($\geq 90\%$ global range in drylands).

Percentages show the proportion of species assessed affected by threats per taxon. Most species are subjected to multiple threats.

COMMENT

There is a lot of potential for looking at biodiversity threats in drylands, and it's good to see that the authors have uncovered major differences in our definitions of drylands. Overall, I had the impression that the authors were attempting to force fit three different work packages into a coherent framework under a discussion of threats, but unfortunately it is not particularly convincing.

Response:

Thank you again for this comment and others. We agree. Therefore, we remove the first section on different definitions of drylands. We have also added an additional analysis evaluating specific threats to dryland species (as described above), with stronger links to land-use changes, protected areas and conservation. Furthermore, we include an additional analysis on the degree of protection of narrow-ranging species as per Reviewer #3's comment. Please see Fig. 4 (below) and Extended Data Fig. 5 (below) and:

Lines 148-164: *“Additionally, we evaluated the protection of dryland species by setting more demanding representation targets of protected area coverage for species with restricted ranges⁴⁷. A 100% protection target was set for those species with ranges smaller than 1000 km² in drylands, and a 10-100% protection target for species with intermediate and widespread ranges above 1000 km². We find that the vast majority of narrow-ranging species in drylands are inadequately protected (Fig. 4 and Supplementary Table S2). For example, 33% of amphibian species in drylands have very narrow ranges (n = 331); of these only 24 species have adequate protection, while 193 species have zero protection. Similarly, 18% of reptiles have very narrow ranges (n = 629); of these only 36 species have adequate protection, 431 species have zero protection. Narrow-ranging birds (n = 65) and mammals (n = 96) are less common, however, of these only two birds and three mammals are adequately protected. Such protection gaps are concerning, as narrow-ranging species are in greatest need of protection due to various threats and are consequently often listed as threatened by the IUCN (e.g., 69% of narrow-ranging bird species are threatened compared to 4% of widespread species; Extended Data Fig. 5). Intermediate and widespread ranging species across taxa are slightly better protected according to representation targets in drylands (amphibians = 18%; birds = 18%; mammals = 16%; reptiles = 16%; Fig. 4), nevertheless most dryland vertebrate species are inadequately protected.”*

Overall, we believe the manuscript now represents a more concise and coherent framework exploring current and future threats to global drylands and biodiversity – by focusing on: **(1) the geographical diversity of drylands (UNEP-drylands only), (2) the poor protection of drylands compared to non-drylands, and subsequent inadequate protection of the vast majority of dryland vertebrate species, (3) the IUCN threat status and current anthropogenic threats to dryland species, and finally (4) the considerable projected land-**

use changes under different future climate change and socioeconomic scenarios threatening the integrity of drylands and survival of dryland species. We hope the reviewer agrees.

[Partially Redacted]

Fig. 4. Protection of amphibian, bird, mammal, and reptile dryland species ($\geq 50\%$ global range in drylands) according to representation targets. Narrow-ranging species = $< 1000 \text{ km}^2$; intermediate = $1000\text{--}250,000 \text{ km}^2$, widespread = $> 250,000 \text{ km}^2$. The protection target for narrow-ranging species is 100% of range size, the protection target for intermediate and widespread species is 10–100% of range size. Scale shows proportion of species with range size protection (IUCN categories I–IV): orange = 0%, blue = 0–10%, green = 10–100%, and dark green = 100% range protection. Percentage values above barplots are the fraction of each taxons' protection target covered in protected areas.

[Partially Redacted]

Extended Data Fig. 5. IUCN threat status of amphibian, bird, mammal, and reptile species in drylands ($\geq 50\%$ global range in drylands) by range size according to representation targets. Narrow-ranging species = $< 1000 \text{ km}^2$; intermediate = $1000\text{-}250,000 \text{ km}^2$, widespread = $> 250,000 \text{ km}^2$. Circles show proportion of IUCN threatened species: dark orange = threatened (IUCN categories CR, EN, VU), dark green = non-threatened (IUCN categories NT, LC), grey = data deficient (IUCN category DD). n = number of species.

RESPONSE TO REVIEWER #3

COMMENT

10 October. First, deepest sympathies with what your country is currently experiencing!

Response:

Our sincerest gratitude. We wish for calm and security on all sides and are thankful for the privilege to conduct conservation science.

COMMENT

I have reviewed Lewin et al.'s manuscript titled "Current and future threats to global drylands". In this manuscript, the authors first show overlaps and differences between four different global maps of dryland. They then proceed to examine how one of these layers (the results from the other three are presented in the supplementary material): 1) overlap with protected areas, 2) represent biodiversity (using amphibians, birds, mammals, and reptiles), and 3) look at how dryland overlaps with projected land cover changes in 2100. As the authors also point out, drylands are often overlooked in conservation and, thus, this paper provides an important (mostly descriptive) contribution to the conservation literature that I believe will be relevant for a broad audience, and which I believe falls very well within the scope of the journal. The manuscript is well written in a clear English and I overall only have a few comments:

Response:

Thank you! We agree that this is a largely descriptive contribution (with some insightful analyses on protected areas, dryland species, and current and future threats to drylands). **Our main objective is to highlight the urgent need for the strategic development of conservation targets and the conservation of biodiversity and ecological systems in drylands in the face of considerable anthropogenic threats.** Due to its broad scope and importance we hope that this research will indeed be relevant to a large and diverse audience, such as within the scope of *Nature Ecology and Evolution*.

COMMENT

I think the authors are missing an opportunity to do more on their analysis comparing the four layers. A more in-depth compare-and-contrast analysis that looks into what environmental and human factors describe the differences would be very interesting. I realize this could likely be its own paper and might be too much for this manuscript (that is likely for the editor and authors to decide), but I did feel that the first part of the results; simply looking at where the four measures overlap or not, left me with more questions than answers – and questions I believe would not be

too hard to answer using additional data on climate, human pressure, land use, habitat classifications etc.

I think the text on the four different layers (ca. lines 74-89) would be much better represented by a table that outlines the main aspects of the four layers. I would also make it much easier to highlight what are the important differences in columns (i.e. source data, resolution, types drylands). I think this would aid the reader and allow for the authors to structure their main points about the data more transparently.

Response:

Following these important comments and similar comments from the other reviewers we removed this first section comparing dryland delineations. We agree that “a more in-depth compare-and-contrast analysis that looks into what environmental and human factors describe the differences would be very interesting” – our intention is to explore this section and these suggestions more thoroughly in a separate manuscript. We now focus only on the UNEP-WCMC drylands dataset as Reviewer #1 suggests. Please see the section titled “Global drylands” where we describe the dataset briefly, and Fig.1:

Lines 66-82: *“We evaluated drylands as classified by the UNEP-WCMC (United Nations Environment World Conservation Monitoring Centre)¹. Drylands are categorized using an aridity index (the ratio of annual precipitation to potential evapotranspiration), with values below 0.65. Drylands are further divided into hyperarid, arid, semiarid, and dry subhumid subtypes (Fig. 1). This drylands designation dataset is commonly referred to by the Convention of Biological Diversity (CBD)³⁴ and United Nations Convention to Combat Desertification (UNCCD)³⁵ for policy and management goals, and often cited in the scientific literature (e.g., ref.⁸). Drylands cover about 42% of earth’s terrestrial surface and encompass diverse and unique regions globally (Fig. 1). Most dryland regions are found in Africa and Asia, which combined encompass 64% of global drylands and also comprise the largest areas of hyperarid and arid subtypes (Extended Data Table 1). Drylands are mostly inclusive of the Deserts & Xeric Shrublands Biome^{10,11}, and also comprise large proportions of other biomes and diverse habitats³ across continents. These range from subhumid zones (e.g., Tropical & Subtropical Grasslands, Savannas & Shrublands) especially in regions of Africa and Australia (Fig. 1 and Extended Data Fig. 1), to Mediterranean regions of Europe, temperate zones (e.g., Temperate Grasslands, Savannas & Shrublands) especially in Asia and North America, and small portions of sub-polar regions (e.g., Boreal Forests/Taiga and Temperate Conifer Forests).”*

COMMENT

Presenting results in a journal that has (for reasons that are beyond my comprehension) chosen to put the methods after the results, requires I think, a bit more indirect introduction of the methods in the results. For example in the sentence starting on line 126, I would suggest the addition of the data (i.e. the World Database on Protected Areas) be included. This could be like: “Using the

World Database on Protected Areas [REF] we show that drylands are considerably less covered by the global network of protected areas compared to non-drylands (Fig. 2a).” This is just a suggestion, but I think the results section, more generally, would benefit from these types of small additions that makes it a little less necessary to flip between the methods and results when reading.

Response:

Done here and in other parts. For example, please see:

Lines 88-91: *“Using the World Database on Protected Areas³⁶ and OpenStreetMap³⁷, we show that only ~12% of total drylands are covered by protected areas (considering all IUCN categories and uncategorized protected areas) – well below global conservation targets to be achieved by 2030 (i.e., 30% coverage of ecologically representative areas)³⁴.”*

Lines 165-166: *“We further explored threat types currently affecting dryland species as designated by the IUCN threat-classification scheme per species (v.3.3; Fig. 5).”*

COMMENT

Figure 2: the description of the white portion of the graph seems misleading in the figure legend. It currently reads “white = all IUCN categories I-VI.”, but isn’t it the sum of the color and the white that is all IUCN categories – the white part alone is only categories V and VI?

Response:

Thank you for this important comment. In fact, the white part represents categories V and VI and also categories ‘Not Applicable’, ‘Not Assigned’ and ‘Not Reported’. We clarify this now in the Methods section and the figure legend:

Lines 121-124: *“Fig. 2. Protected areas in drylands. a, Proportion of protected areas by dryland subtype compared to total non-drylands. b, Proportion of protected areas in drylands by continent compared to non-drylands. Coloured bars = IUCN categories I-IV; white bars = IUCN categories V-VI and uncategorized (i.e., ‘Not Applicable’, ‘Not Assigned’ and ‘Not Reported’).”*

Lines 488-495: *“We conducted this protected area analysis at two levels: (1) with protected areas defined by International Union of Conservation (IUCN) management categories ‘I-IV’, which are well suited for protecting biological diversity and restrict human activities⁴¹, and (2) for all categories of protected areas – using IUCN management categories of protected areas ‘I-IV’ and categories ‘V and VI’ (permitting resource extraction and mixed land uses), including protected areas that lack an explicit IUCN management category for various reasons (i.e., categorized as ‘Not Applicable’, ‘Not Assigned’ and ‘Not Reported’) – herein ‘uncategorized’.”*

COMMENT

The use of thresholds of protection (i.e. 10%) is a bit crude and doesn't include long established approaches to integrate the total range-size of a species when assessing adequacy of protection. There is nothing wrong in what the authors do, but it would be more informative and align better with conventions of how this is reported, if the authors used a measure of adequacy rather than an arbitrary 10% threshold. I would suggest the authors consult Rodrigues, et al. 1, to see the methods used there, which would allow the authors to use more meaningful measures of 1) the proportion of species with no protection, 2) the proportion with adequate protection, and 3) the proportion with some but not adequate protection. It might also be worth looking at Butchart, et al. 2 for a more recent version.

Response:

Thank you for this comment. We now add this analysis on protection targets considering the range size of species. The results are very insightful especially to highlight the inadequate protection of species with very narrow-ranging species. These complement our results using arbitrary thresholds showing that the vast majority of dryland species do not have adequate protection of their total range sizes in drylands. Please see:

Lines 148-164: *“Additionally, we evaluated the protection of dryland species by setting more demanding representation targets of protected area coverage for species with restricted ranges⁴⁷. A 100% protection target was set for those species with ranges smaller than 1000 km² in drylands, and a 10-100% protection target for species with intermediate and widespread ranges above 1000 km². We find that the vast majority of narrow-ranging species in drylands are inadequately protected (Fig. 4 and Supplementary Table S2). For example, 33% of amphibian species in drylands have very narrow ranges (n = 331); of these only 24 species have adequate protection, while 193 species have zero protection. Similarly, 18% of reptiles have very narrow ranges (n = 629); of these only 36 species have adequate protection, 431 species have zero protection. Narrow-ranging birds (n = 65) and mammals (n = 96) are less common, however, of these only two birds and three mammals are adequately protected. Such protection gaps are concerning, as narrow-ranging species are in greatest need of protection due to various threats and are consequently often listed as threatened by the IUCN (e.g., 69% of narrow-ranging bird species are threatened compared to 4% of widespread species; Extended Data Fig. 5). Intermediate and widespread ranging species across taxa are slightly better protected according to representation targets in drylands (amphibians = 18%; birds = 18%; mammals = 16%; reptiles = 16%; Fig. 4), nevertheless most dryland vertebrate species are inadequately protected.”*

Please see also Fig. 4 (below) and Extended Data Fig. 5 (below) and Supplementary Table S2, and in Methods:

Lines 522-528: *“Additionally, we evaluated the protection of dryland species by setting representation targets of protection for species according to range size (Fig. 4). A 100% protection target was set for species with ranges <1000 km² in drylands, and a 10-100% protection target for species with intermediate ranges (1000-250,000 km²) and widespread ranges (>250,000 km²). We categorized range protection as follows: 0% protection (<0.1% of range size protected), 0-10% (0.1-9.9%), 10-100% protection (9.9-99%), 100% protection (>99%). We repeated these analyses at the continental level (Supplementary Table S2).”*

[Partially Redacted]

Fig. 4. Protection of amphibian, bird, mammal, and reptile dryland species ($\geq 50\%$ global range in drylands) according to representation targets. Narrow-ranging species = $< 1000 \text{ km}^2$; intermediate = $1000\text{-}250,000 \text{ km}^2$, widespread = $> 250,000 \text{ km}^2$. The protection target for narrow-ranging species is 100% of range size, the protection target for intermediate and widespread species is 10-100% of range size. Scale shows proportion of species with range size protection (IUCN categories I-IV): orange = 0%, blue = 0-10%, green = 10-100%, and dark green = 100% range protection. Percentage values above barplots are the fraction of each taxons' protection target covered in protected areas.

[Partially Redacted]

Extended Data Fig. 5. IUCN threat status of amphibian, bird, mammal, and reptile species in drylands ($\geq 50\%$ global range in drylands) by range size according to representation targets. Narrow ranging species = $< 1000 \text{ km}^2$; intermediate = $1000\text{-}250,000 \text{ km}^2$, widespread = $> 250,000 \text{ km}^2$. Circles show proportion of IUCN threatened species: dark orange = threatened (IUCN categories CR, EN, VU), dark green = non-threatened (IUCN categories NT, LC), grey = data deficient (IUCN category DD). n = number of species.

COMMENT

In the use of the WDPA, it's important to realize that the database has some major gaps. Some countries don't allow for their data to be displayed and downloaded. This can change with the specific version, but from 2017/18 this has been China and more recently (but probably before 2022) India and a few other countries. If all that has been done in terms of the protected area results is some filtering based on IUCN management categories the results might be misleading for reasons that simply has to do with what data has been used. The authors will need to check for any systematic gaps in the data and incorporate this in the results. This should be possible to see by simply visually inspecting the data on a map. Alternatively, this should be reported in the notes to the specific version downloaded. This likely means re-running all protected area analysis.

Response:

The reviewer is absolutely correct. We have systematically reviewed these gaps as suggested above. As a result, we amend the database for China, and in addition for gaps in reporting of protected areas in India and Turkey. At the same time, we used the most recent update of the WDPA database in November 2023, and therefore rerun all analyses accordingly as the reviewer suggests. The general trends in the degree of protection of drylands compared to non-drylands, and the protection of dryland species are similar to those initially reported, but of course we revise and report these updated results accordingly throughout the manuscript.

Please see Supplementary Fig. S1 (below), and:

Lines 481-488: *“To assess the degree of protected area coverage of drylands (i.e., proportion dryland area protected), we overlaid drylands with the World Database on Protected Areas³⁶ (data downloaded on November 2023 from www.protectedplanet.net/). Some regions lacked updated WDPA data (i.e., China), therefore for China we merged WDPA data from 2016 (when data was last reported fully) with 2023. Other regions did not provide complete WDPA data (i.e., Turkey and India), therefore for Turkey and India we compiled data on protected areas using OpenStreetMap³⁷ (data downloaded on November 2023 from www.openstreetmap.org/; see Supplementary Fig. S1 for comparisons to WDPA 2023 data).”*

Supplementary Fig. S1. Protected areas for regions lacking updated World Database on Protected Areas (WDPA) data: China (a), India (b), and Turkey (c). IUCN categories I-VI and uncategorized (i.e., 'Not Applicable', 'Not Assigned' and 'Not Reported').

COMMENT

Minor comments

Lines: 71-72: Not sure what this sentence means

Line 72: should “compared” be present tense (i.e. “compare”)

Response:

The section has now been removed.

Line 137: strictness (i.e. strictly for biodiversity) in relation to the IUCN categories is a complicated discussion. For example, category VI have in many cases been shown to outperform other “lower” level/numbered categories. And indeed, in particular category IV, but also III, and II allows for other considerations than biodiversity. Though I would personally never divide the IUCN categories into I-IV and V-VI, I think the division can be accepted, but the language around it needs to be refined.

Response:

We agree, and refine the language slightly:

Lines 100-102: *“Considering protected areas managed mainly for science, wilderness protection, habitat and species conservation (i.e., IUCN categories I-IV)⁴¹ – a considerably smaller proportion of drylands are protected.”*

We also acknowledge the potential effectiveness of other forms of protection:

Lines 106-108: *“In Africa and Asia, protected areas including community and indigenous conserved areas without an IUCN category may be under-reported⁴², which might actively support effective biodiversity conservation⁴³.”*

Lines 259-263: *“At the same time, drylands provide unique opportunities and significant scope for achieving conservation and biodiversity goals, especially in Africa and Asia, including the expansion of protected area networks and incorporating stricter forms of protection, while recognizing community-based approaches to conservation^{72,73} and other management strategies^{50,74}.”*

Lines 213-214 and Lines 224-225: in the first case the authors write: “... (SSP2-4.5), 100% of drylands are projected to be converted to human land uses” and in the second “... (SSP5-8.5), most natural drylands are projected to undergo some degree of conversion to human land-uses”. I

am a little confused as to how 100% was converted under the less depressing scenario and “only” most was projected to “undergo some degree of conversion”. It might just be the authors trying to not make the text too “dry”/repetitive, but it actually opens up room for some misunderstandings.

Response:

This is a valid comment, we clarify this discrepancy now to avoid confusion:

Lines 233-239: *“Similarly, in a projected extreme scenario characterized by rapid and resource-intensive development and high emissions (SSP5-8.5), most natural drylands are projected to undergo some degree of conversion to human land uses (10.4 M km² additional natural drylands will be converted; Fig. 6d). However, these conversion rates are slightly lower than the ‘Middle Road Scenario’⁵⁹ (SSP2-4.5; Fig 6c), mostly due to extreme climate and water restrictions limiting cropland, pasture and rangeland expansion in some dryland regions (Extended Data Fig. 10), coupled with declines in global population⁶⁹.”*

Line 422: I just need to be sure, you were working at a 0.002 km² resolution in a global analysis? Or what is this?

Response:

This is correct, but we omit this section now.

Lines 436-442: it’s not entirely clear to me from the description here how protected areas that was not categorized as one of the six/seven IUCN categories were treated. I don’t have strong opinions about how they should be used (or not used) but clarity about what was done is important.

Response:

We clarified this now in the Methods (as explained above) and all relevant figure legends in the Main Text and Extended Data:

Lines 488-495: *“We conducted this protected area analysis at two levels: (1) with protected areas defined by International Union of Conservation (IUCN) management categories ‘I-IV’, which are well suited for protecting biological diversity and restrict human activities⁴¹, and (2) for all categories of protected areas – using IUCN management categories of protected areas ‘I-IV’ and categories ‘V and VI’ (permitting resource extraction and mixed land uses), including protected areas that lack an explicit IUCN management category for various reasons (i.e., categorized as ‘Not Applicable’, ‘Not Assigned’ and ‘Not Reported’) – herein ‘uncategorized’.”*

Thank you again for your comments!

Decision Letter, first revision:

17th April 2024

Dear Dr. Lewin,

Thank you for your patience while your revised manuscript "Current and future threats to global drylands" was under review (NATECOLEVOL-23092141A). On the basis of the reviewers' comments, my colleagues and I will be happy, in principle, to publish it in Nature Ecology & Evolution, pending minor revisions to address the reviewers' final requests and to comply with our editorial and formatting guidelines.

We are now performing detailed checks on your paper and will send you a checklist detailing our editorial and formatting requirements in about a week. **Please do not upload the final materials or make any revisions until you receive this additional information from us.**

[REDACTED]

Reviewer #1 (Remarks to the Author):

I feel that the authors have done an adequate job in responding to the extensive reviewer comments, and would like to thank the authors for considering all my comments and explaining the changes made in such a detailed way.

I don't have additional comments to make other than the suggestion to include as supplementary material the additional analyses shown in Response Figures 1-3 as I feel that doing so would make the ms more robust and further convince critical readers like me.

I also would recommend to slightly edit the final paragraph of the introduction as follows. I would split the first sentence into two. In the second part the authors could expand a little bit how such a systematic approach can help management and conservation actions. I would also expand a little bit more the objectives perhaps adding another 1-2 sentences to highlight their relevance

Congratulations to the authors, without any doubt this is an interesting and valuable paper and will make a substantial contribution to the dryland literature.

Fernando T. Maestre

Reviewer #2 (Remarks to the Author):

The authors have done a great job at addressing not only my concerns, but the concerns of the other reviewers. I find that the manuscript is now more coherent, and presents a much more unambiguous story. I have no further comments. Well done

David Eldridge

Reviewer #3 (Remarks to the Author):

I have been through the new version of the manuscript - focusing in particular on the comments I made to the first version (however do to the considerable re-write and re-analysis I have also been through the main manuscript and looked at the comments of the other reviewers.

I think the authors have done a good job of addressing the reviewers' comment, and based on that I have no major comments to the new version. this represents an interesting analysis and the authors have clearly taken a very constructive approach to the reviewers' comment - one that has led to a substantial amount of re-analysis that have strengthened the manuscript and clarified important aspects.

Our ref: NATECOLEVOL-23092141A

23rd April 2024

Dear Dr. Lewin,

Thank you for your patience as we've prepared the guidelines for final submission of your Nature Ecology & Evolution manuscript, "Current and future threats to global drylands" (NATECOLEVOL-23092141A). Please carefully follow the step-by-step instructions provided in the attached file, and add a response in each row of the table to indicate the changes that you have made. Please also check and comment on any additional marked-up edits we have proposed within the text. Ensuring that each point is addressed will help to ensure that your revised manuscript can be swiftly handed over to our production team.

****We would like to start working on your revised paper, with all of the requested files and forms, as soon as possible (preferably within two weeks). Please get in contact with us immediately if you anticipate it taking more than two weeks to submit these revised files.****

In recognition of the time and expertise our reviewers provide to Nature Ecology & Evolution's editorial process, we would like to formally acknowledge their contribution to the external peer review of your manuscript entitled "Current and future threats to global drylands". For those reviewers who give their assent, we will be publishing their names alongside the published article.

Nature Ecology & Evolution offers a Transparent Peer Review option for new original research manuscripts submitted after December 1st, 2019. As part of this initiative, we encourage our authors to support increased transparency into the peer review process by agreeing to have the reviewer comments, author rebuttal letters, and editorial decision letters published as a Supplementary item. When you submit your final files please clearly state in your cover letter whether or not you would like to participate in this initiative. Please note that failure to state your preference will result in delays in accepting your manuscript for publication.

Cover suggestions

We welcome submissions of artwork for consideration for our cover. For more information, please see our guide for cover artwork.

Nature Ecology & Evolution has now transitioned to a unified Rights Collection system which will allow our Author Services team to quickly and easily collect the rights and permissions required to publish your work. Approximately 10 days after your paper is formally accepted, you will receive an email in providing you with a link to complete the grant of rights. If your paper is eligible for Open Access, our Author Services team will also be in touch regarding any additional information that may be required to arrange payment for your article.

Please note that *Nature Ecology & Evolution* is a Transformative Journal (TJ). Authors may publish their research with us through the traditional subscription access route or make their paper immediately open access through payment of an article-processing charge (APC). Authors will not be required to make a final decision about access to their article until it has been accepted. Find out more about Transformative Journals

Authors may need to take specific actions to achieve compliance with funder and institutional open access mandates. If your research is supported by a funder that requires immediate open access (e.g. according to Plan S principles) then you should select the gold OA route, and we will direct you to the compliant route where possible. For authors selecting the subscription publication route, the journal's standard licensing terms will need to be accepted, including [a href="https://www.nature.com/nature-portfolio/editorial-policies/self-archiving-and-license-to-publish"](https://www.nature.com/nature-portfolio/editorial-policies/self-archiving-and-license-to-publish). Those licensing terms will supersede any other terms that the author or any third party may assert apply to any version of the manuscript.

For information regarding our different publishing models please see our Transformative

Journals page. If you have any questions about costs, Open Access requirements, or our legal forms, please contact ASJournals@springernature.com.

[REDACTED]

[REDACTED]

Reviewer #1:

Remarks to the Author:

I feel that the authors have done an adequate job in responding to the extensive reviewer comments, and would like to thank the authors for considering all my comments and explaining the changes made in such a detailed way.

I don't have additional comments to make other than the suggestion to include as supplementary material the additional analyses shown in Response Figures 1-3 as I feel that doing so would make the ms more robust and further convince critical readers like me.

I also would recommend to slightly edit the final paragraph of the introduction as follows. I would split the first sentence into two. In the second part the authors could expand a little bit how such a systematic approach can help management and conservation actions. I would also expand a little bit more the objectives perhaps adding another 1-2 sentences to highlight their relevance

Congratulations to the authors, without any doubt this is an interesting and valuable paper and will make a substantial contribution to the dryland literature.

Fernando T. Maestre

Reviewer #2:

Remarks to the Author:

The authors have done a great job at addressing not only my concerns, but the concerns of the other reviewers. I find that the manuscript is now more coherent, and presents a much more unambiguous story. I have no further comments. Well done

David Eldridge

Reviewer #3:

Remarks to the Author:

I have been through the new version of the manuscript - focusing in particular on the comments I made to the first version (however do to the considerable re-write and re-analysis I have also been through the main manuscript and looked at the comments of the other reviewers.

I think the authors have done a good job of addressing the reviewers' comment, and based on that I have no major comments to the new version. this represents an interesting analysis and the authors have clearly taken a very constructive approach to the reviewers' comment - one that has led to a substantial amount of re-analysis that have strengthened the manuscript and clarified important aspects.

Author Rebuttal, first revision:

REVIEWER #1

Remark to the Author:

I feel that the authors have done an adequate job in responding to the extensive reviewer comments, and would like to thank the authors for considering all my comments and explaining the changes made in such a detailed way.

I don't have additional comments to make other than the suggestion to include as supplementary material the additional analyses shown in Response Figures 1-3 as I feel that doing so would make the ms more robust and further convince critical readers like me.

Response:

We thank the reviewer for the positive and constructive evaluation of our work.

We now include this additional analysis (Response Fig. 1) as a supplementary figure (Supplementary Figure 5) in the Supplementary Information (and included here below) – comparing the LUH2 dataset we applied to drylands with a recently published dataset – the Future Land Use Simulation model (FLUS) by Chen et al (2022).

However, as response Figs. 2 & 3 are not our own original analyses but rather figures from already published manuscripts, we prefer to cite these appropriately in the main text as below. If the editor disagrees with this approach, then we would gladly also incorporate these figures in the Supplementary Information as suggested by the reviewer.

Lines 331-341: *“Recently released datasets at higher resolutions – see ref. ⁷⁷ (e.g., Fig. 10), ref. ⁷⁸ (e.g., Fig. 4), and ref. ⁷⁹ – correlate strongly with LUH2 spatially for comparable land classifications, however these datasets may have several limitations based on data availability yielding potential errors in future land projections. Also, these datasets do not distinguish between grazing areas (i.e., rangeland, grassland and pasture regions), which have particular relevance and distinctions in drylands. We compared the LUH2 dataset applied here to drylands with a recently published dataset – the Future Land Use Simulation model (FLUS)⁷⁸. When comparing global values of total land areas (10⁶ km) of similar land types under projected future scenarios, we find that the general trajectories are similar to LUH2 in drylands (see Supplementary Figure 5). Therefore, we use LUH2 – widely applied in spatially explicit global land-use analyses (e.g., refs. ^{80–82}).”*

References:

77. Zhang, T., Cheng, C. & Wu, X. Mapping the spatial heterogeneity of global land use and land cover from 2020 to 2100 at a 1 km resolution. *Sci Data* 10, (2023).
78. Chen, G., Li, X. & Liu, X. Global land projection based on plant functional types with a 1-km resolution under socio-climatic scenarios. *Sci Data* 9, (2022).
79. Chen, M. et al. Global land use for 2015–2100 at 0.05° resolution under diverse socioeconomic and climate scenarios. *Sci Data* 7, (2020).

Supplementary Figure 5. Plots showing changes in land area for major land types in drylands from datasets LUH2 and FLUS (from Chen et al., 2022)⁷⁸ under SSP scenarios, showing comparable trends for the major land types considered. Lines show differences in land area (10⁶ km) between years 2015 and 2100 for scenario SSP1-2.6 (2100) in blue, SSP2-4.5 (2100) in green, and SSP5-8.5 (2100) in red.

Remark to the Author:

I also would recommend to slightly edit the final paragraph of the introduction as follows. I would split the first sentence into two. In the second part the authors could expand a little bit how such a systematic approach can help management and conservation actions. I would also expand a little bit more the objectives perhaps adding another 1-2 sentences to highlight their relevance.

Response:

Please see below where we incorporate the reviewer's comments in the final paragraph of the Introduction.

Lines 56-63: "Consequently, there is an urgent need for a systematic approach to evaluate current and future threats to drylands and their vertebrate biodiversity. Such an approach will have important implications for guiding land management and conservation strategies in drylands by identifying broad-scale conservation priorities³³, and increasing conservation targets across all ecosystems. Here, we assess the degree of protected area coverage in drylands and dryland subtypes compared to non-drylands, the current status of anthropogenic threats to dryland vertebrate biodiversity and highlight important protection gaps, and the impact that projected future human land-use pressures will have on drylands under different climate and socioeconomic pathways."

Remark to the Author:

Congratulations to the authors, without any doubt this is an interesting and valuable paper and will make a substantial contribution to the dryland literature.

Response:

We thank the reviewer very much again for their comments.

REVIEWER #2:**Remarks to the Author:**

The authors have done a great job at addressing not only my concerns, but the concerns of the other reviewers. I find that the manuscript is now more coherent, and presents a much more unambiguous story. I have no further comments. Well done.

Response:

We thank the reviewer very much.

REVIEWER #3:

Remarks to the Author:

I have been through the new version of the manuscript - focusing in particular on the comments I made to the first version (however do to the considerable re-write and re-analysis I have also been through the main manuscript and looked at the comments of the other reviewers.

I think the authors have done a good job of addressing the reviewers' comments, and based on that I have no major comments to the new version. This represents an interesting analysis and the authors have clearly taken a very constructive approach to the reviewers' comments – one that has led to a substantial amount of re-analysis that has strengthened the manuscript and clarified important aspects.

Response:

We thank the reviewer very much.

Final Decision Letter:

Dear Dr Lewin,

We are pleased to inform you that your Article entitled "Global evaluation of current and future threats to drylands and their vertebrate biodiversity" has now been accepted for publication in *Nature Ecology & Evolution*.

Over the next few weeks, your paper will be copyedited to ensure that it conforms to *Nature Ecology and Evolution* style. Once your paper is typeset, you will receive an email with a link to choose the appropriate publishing options for your paper and our Author Services team will be in touch regarding any additional information that may be required

Due to the importance of these deadlines, we ask you please us know now whether you will be difficult to contact over the next month. If this is the case, we ask you provide us with the contact information (email, phone and fax) of someone who will be able to check the proofs on your behalf, and who will be available to address any last-minute problems . Once your paper has been scheduled for online publication, the Nature press office will be in touch to confirm the details.

Acceptance of your manuscript is conditional on all authors' agreement with our publication policies (see www.nature.com/authors/policies/index.html). In particular your manuscript must not be published elsewhere and there must be no announcement of the work to any media outlet until the publication date (the day on which it is uploaded onto our web site).

Please note that *Nature Ecology & Evolution* is a Transformative Journal (TJ). Authors may publish

their research with us through the traditional subscription access route or make their paper immediately open access through payment of an article-processing charge (APC). Authors will not be required to make a final decision about access to their article until it has been accepted. Find out more about Transformative Journals

Authors may need to take specific actions to achieve compliance with funder and institutional open access mandates. If your research is supported by a funder that requires immediate open access (e.g. according to Plan S principles) then you should select the gold OA route, and we will direct you to the compliant route where possible. For authors selecting the subscription publication route, the journal's standard licensing terms will need to be accepted, including [a href="https://www.nature.com/nature-portfolio/editorial-policies/self-archiving-and-license-to-publish](https://www.nature.com/nature-portfolio/editorial-policies/self-archiving-and-license-to-publish). Those licensing terms will supersede any other terms that the author or any third party may assert apply to any version of the manuscript.

We welcome the submission of potential cover material (including a short caption of around 40 words) related to your manuscript; suggestions should be sent to Nature Ecology & Evolution as electronic files (the image should be 300 dpi at 210 x 297 mm in either TIFF or JPEG format). Please note that such pictures should be selected more for their aesthetic appeal than for their scientific content, and that colour images work better than black and white or grayscale images. Please do not try to design a cover with the Nature Ecology & Evolution logo etc., and please do not submit composites of images related to your work. I am sure you will understand that we cannot make any promise as to whether any of your suggestions might be selected for the cover of the journal.

You can generate the link yourself when you receive your article DOI by entering it here: <http://authors.springernature.com/share>.

[REDACTED]

P.S. Click on the following link if you would like to recommend Nature Ecology & Evolution to your librarian <http://www.nature.com/subscriptions/recommend.html#forms>

nature portfolio

** Visit the Springer Nature Editorial and Publishing website at www.springernature.com/editorial-and-publishing-jobs for more information about our career opportunities. If you have any questions please click here.**